# Beta-containing bivalent SARS-CoV-2 protein vaccine elicits durable broad neutralization in macaques and protection in hamsters

Catherine Berry [1,8], Vincent Pavot [1,8], Natalie G. Anosova[2], Michael Kishko [2], Lu Li[2], Tim Tibbitts[2], Alice Raillard[1], Sylviane Gautheron [1], Sheila Cummings[3], Dinesh S. Bangari [3], Swagata Kar[4], Caroline Atyeo[5], Yixiang Deng[5], Galit Alter [5], Cindy Gutzeit[6], Marguerite Koutsoukos [7], Roman M. Chicz[2] & Valerie Lecouturier [1✉]

## Abstract

**Background** Since the beginning of the COVID-19 pandemic, several variants of concern (VOC) have emerged for which there is evidence of an increase in transmissibility, more severe disease, and/or reduced vaccine effectiveness. Effective COVID-19 vaccine strategies are required to achieve broad protective immunity against current and future VOC.

**Methods** We conducted immunogenicity and challenge studies in macaques and hamsters using a bivalent recombinant vaccine formulation containing the SARS-CoV-2 prefusion-stabilized Spike trimers of the ancestral D614 and the variant Beta strains with AS03 adjuvant (CoV2 preS dTM-AS03) in a primary immunization setting.

**Results** We show that a primary immunization with the bivalent CoV2 preS dTM-AS03 elicits broader and durable (1 year) neutralizing antibody responses against VOC including Omicron BA.1 and BA.4/5, and SARS-CoV-1 as compared to the ancestral D614 or Beta variant monovalent vaccines in naïve non-human primates. In addition, the bivalent formulation confers protection against viral challenge with SARS-CoV-2 prototype D614G strain as well as Alpha and Beta variant strains in hamsters.

**Conclusions** Our findings demonstrate the potential of a Beta-containing bivalent CoV2 preS dTM-AS03 formulation to provide broad and durable immunogenicity, as well as protection against VOC in naïve populations.

## Plain language summary

SARS-CoV-2 has changed over time, resulting in different forms of the virus called variants. These variants compromise the protection offered by the COVID-19 vaccines, which trigger an immune response against the viral Spike protein that allows the virus to attach and infect human cells, since their spike proteins are different. Here, we developed and tested a vaccine containing two different Spike proteins, one from the original Wuhan strain and another from the Beta variant. In macaques, the vaccine leads to the production of antibodies able to stop all variants tested from infecting human cells, including Omicron, with stable levels over one year. In hamsters, the vaccine protected against infection with the ancestral virus and the Alpha and Beta variants. Our findings have important implications for vaccine control of existing and future SARS-CoV-2 variants of concern.

[1] Sanofi, Vaccines R&D, Marcy l'Etoile, France. [2] Sanofi, Vaccines R&D, Cambridge, MA, USA. [3] Sanofi, Framingham, MA, USA. [4] BIOQUAL Inc, Rockville, MD, USA. [5] Ragon Institute of MGH, MIT, and Harvard, Cambridge, MA, USA. [6] GSK, Rixensart, Belgium. [7] GSK, Wavre, Belgium. [8] These authors contributed equally: Catherine Berry, Vincent Pavot. ✉email: valerie.lecouturier@sanofi.com

Severe acute respiratory syndrome coronavirus 2 (SARS-CoV-2), responsible for coronavirus disease (COVID-19), emerged in late 2019. During the first six months of the pandemic, a variant with a single mutation in the Spike antigen (D614G), replaced the circulating ancestral Wuhan strain (D614). However, the relative genetic stability ended after the first year when variants of concern (VOC) displaying multiple mutations in the Spike antigen, successively displaced the preceding strain causing multiple waves of infections worldwide. Several VOC have been identified (Alpha, Beta, Gamma, Delta, and Omicron BA.1), along with highly diverse subvariants of Omicron, some of which have been classified as new VOC by the European Centre for Disease Prevention and Control (BA.4 and BA.5)[1]. SARS-CoV-2 evolution continues at a higher speed than previously anticipated and is now mainly driven by immune escape as demonstrated by the high incidence of Omicron symptomatic disease in vaccinated individuals[2]. Although multiple vaccines have been approved for use in a record time, all the authorized vaccines were originally designed to target the ancestral D614 strain[3,4].

It is now well established that SARS-CoV-2 is endemic, with unpredictable severity[5]. Thus, there is an urgent unmet need for optimized variant-proof COVID-19 vaccines to provide broad and durable protection against existing and future variants.

Because of the continuous fast replacement of BA.1 by other Omicron sub-lineages (BA.2, BA.2.12.1, BA.4, BA.5, BQ.1, BQ.1.1 …) showing similar to higher immune escape and transmission rates than BA.1[6–9], the current vaccine strategies chasing each new variants need to be re-evaluated, for booster doses as well as for primary immunization[10,11]. Indeed, next-generation vaccine strategies need to address efficient protection for the unvaccinated and naïve population, such as young children at risk against future circulating VOC[12].

We recently showed that soluble prefusion-stabilized Spike trimers (CoV2 preS dTM) based on the D614 sequence formulated with the AS03 adjuvant elicited a favorable safety profile and high immunogenicity against the ancestral strain in naïve adults[13]. In addition, we demonstrated that, in primed macaques, one booster dose of CoV2 preS dTM-AS03 (D614 or Beta) enhanced neutralizing antibodies (nAbs) against the ancestral virus and extended the neutralization to multiple VOC (Alpha, Beta, Gamma, Delta, and Omicron) and SARS-CoV-1[14,15]. Preliminary data from two clinical trials assessing CoV2 preS dTM-AS03 as a booster in COVID-19 vaccine-primed individuals confirmed these results and further suggested the superiority of the Beta monovalent vaccine formulations compared to D614 monovalent mRNA or subunit vaccines[16,17].

Here, we report the immunogenicity and protection of a bivalent recombinant vaccine formulation containing the prefusion-stabilized Spike trimers of the ancestral D614 and the Beta (B.1.351) strains in a primary immunization setting in macaques and hamsters. The Beta Spike sequence was selected based on its significant immune escape to the prototype vaccine-induced neutralizing antibodies mainly attributed to the three RBD mutations (K417N, E484K, and Y501N)[18–21]. Naïve cynomolgus macaques and hamsters were vaccinated with monovalent (D614 or Beta) or bivalent (D614 + Beta) CoV2 preS dTM-AS03 vaccine candidates to assess the neutralizing antibody responses against vaccine-homologous strains, a wide panel of VOC including Omicron subvariants and SARS-CoV-1 as well as memory B cell and durability of the neutralizing antibody responses. The protection conferred by the Beta-containing vaccine formulations was evaluated in vaccinated hamsters after a challenge with the prototype D614G, Alpha, or Beta variants.

Our data show that primary immunization with the ancestral/Beta bivalent CoV2 preS dTM-AS03 vaccine candidate elicits durable neutralizing antibody responses for up to 1 year, with the broadest coverage against SARS-CoV-2 VOC including Omicron BA.1 and BA.4/5 as well as against SARS-CoV-1 in naïve non-human primates (NHPs), and confers protection against prototype D614G, Alpha, or Beta variant challenge in hamsters.

## Methods

**Vaccines.** Vaccine candidates CoV2 preS dTM D614 or Beta with AS03-adjuvant and formulated as monovalent or bivalent vaccines were described previously in refs. [14,22].

Briefly, CoV2 preS dTM vaccine candidates consist of a stabilized prefusion trimeric recombinant SARS-CoV-2 S protein. The CoV2 preS dTM (D614) and Beta were designed based on the Wuhan YP_009724390.1 and B.1.351 sequences (GISAID Accession EPI_ISL_1048524), respectively, with two prolines in the S2 domain, mutation of the furin cleavage site and replacement of the transmembrane region by the T4 foldon trimerization domain. The CoV2 preS dTM D614 or Beta were produced using a Sanofi proprietary cell culture technology based on the insect cell-baculovirus expression system. The CoV2 preS dTM D614 or Beta are formulated with the AS03 adjuvant from GSK.

**Viruses and cells.** SARS-CoV-2 viral stocks used for the challenge were prepared at Bioqual from seeds obtained from Biodefense and Emerging Infections Research Resources Repository (BEI Resources). SARS-CoV-2 USA/NY-PV08449/2020 (D614G) stock was derived from BEI seed stock NR-53515 and assigned Lot # 091620-230. The stock titer is $9.8 \times 10^4$ TCID$_{50}$/mL (in VeroE6) and $5 \times 10^7$ TCID$_{50}$/mL in Vero TMPRSS2. SARS-CoV-2 Alpha variant (B.1.1.7) stock was derived from BEI seed stock NR-54011 and assigned Lot # 012921-1230. The stock titer is $1.58 \times 10^7$ TCID$_{50}$/mL and $1.38 \times 10^6$ PFU/mL in Vero TMPRSS2. SARS-CoV-2 Beta variant (B.1.351) stock was derived from BEI seed stock NR-54974 and assigned Lot # 030621-750. The stock titer was determined in VeroE6 and Vero TMPRSS2 cells via both TCID$_{50}$ and Plaque assay. The titer is $5 \times 10^5$ TCID$_{50}$/mL in VeroE6 and $1.99 \times 10^8$ TCID$_{50}$/mL in Vero TMPRSS2.

Reporter virus particles (RVPs)-GFP used in the lentivirus-based pseudovirus neutralization assay were obtained from Integral Molecular (Table 1).

293T-hsACE2 clonal cells (Integral Molecular, Cat# C-HA102) used for the lentivirus-based pseudovirus neutralization assay were obtained from Integral Molecular and grown according to the manufacturer's instructions.

Vero TMPRSS2 cells (obtained from Adrian Creanga, Vaccine Research Center-NIAID) were grown in Dulbecco's Modified Eagle Medium (DMEM) + 10% fetal bovine serum (FBS) + Gentamicin.

**Animals and study design.** Animal experiments were conducted in compliance with all pertinent US National Institutes of Health regulations according to approved animal protocols from the Institutional Animal Care and Use Committee (IACUC) at the research facilities. The NHP study was performed at the University of Louisiana at Lafayette New Iberia Research Center and the hamster study was performed at Bioqual.

Cynomolgus macaques, aged 2–8 years, were randomized based on sex, age, and weight. Groups were composed of six animals, including two or three females and four or three males. Macaques were vaccinated on D0 and D21 by intramuscular route into the deltoid muscle with different vaccine formulations (with AS03 adjuvant): monovalent ancestral Spike D614 (5 or 10 µg), monovalent Beta variant Spike (5 or 10 µg) or bivalent D614 + Beta (5 µg + 5 µg) (Fig. 1). Two other groups received the

**Table 1 Description of the reporter virus particles (RVPs) used in the pseudovirus neutralization assay.**

| Pango Lineage | WHO Name | Strain (Nextstrain) | Sequence source | Catalog number | Lot | Mutations relative to Wuhan D614 |
|---|---|---|---|---|---|---|
| A | N/A | Wuhan (reference sequence) – D614 | GenBank QHD43416.1 | RVP-701 | CG-113A | - |
| B.1 | N/A | D614G | - | RVP-702G | CG-129A | D614G |
| B.1.1.7 | Alpha | 20I/S:501Y.V1 | QQH18545.1 | RVP-706G | CG-135A | ΔH69/V70, ΔY144, N501Y, A570D, D614G, P681H, T716I, S982A, D1118H |
| B.1.351 | Beta | 20H/S:501Y.V2 | Tegally et al. 2020 | RVP-724G | CG-180A | L18F, D80A, D215G, ΔL242/A243/L244, R246I, K417N, E484K, N501Y, D614G, A701V |
| P.1 or B.1.128 | Gamma | 20J/S:501Y.V3 | QQX12069.1 | RVP-708G | CG-160A | L18F, T20N, P26S, D138Y, R190S, K417T, E484K, N501Y, D614G, H655Y, T1027I, V1176 |
| B.1.617.2 | Delta | 21A/S:478K | cov-lineages.org | Custom | CG-233A | T19R, G142D, E156G, ΔF157/R158, L452R, T478K, D614G, P681R, D950N |
| B.1.621 | Mu | 21H | QXK87114.1 | RVP-767G | CG-273A | T95I, Y144T, Y145S, ins146N, R346K, E484K, N501Y, P681H, D614G |
| BA.1 | Omicron BA.1 | 21K | EPI_ISL_6841980 | RVP-768G | CG-296A | A67V, Δ69-70, T95I, G142D/Δ143-145, Δ211/L212I, ins214EPE, G339D, S371L, S373P, S375F, K417N, N440K, G446S, S477N, T478K, E484A, Q493R, G496S, Q498R, N501Y, Y505H, T547K, D614G, H655Y, N679K, P681H, N764K, D796Y, N856K, Q954H, N969K, L981F |
| BA.4/5 | Omicron BA.4/5 | 22B | UTO31503.1 | RVP-774G | CG-352A | T19I, LPPA24S, HV69del, G142D, V213G, G339D, S371F, S373P, S375F, T376A, D405N, R408S, K417N, N440K, L452R, S477N, T478K, E484A, F486V, Q498R, N501Y, Y505H, D614G, H655Y, N679K, P681H, N764K, D796Y, Q954H, N969K |
| NA | NA | SARS-CoV-1 | P59594.1 | RVP-801G | SG-115B | 28% differences |

N/A not applicable.

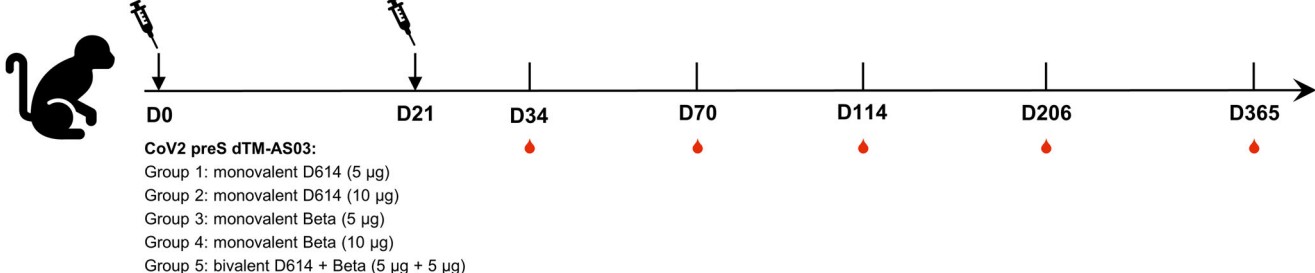

**Fig. 1 Non-human primate study schema.** Five groups of six cynomolgus macaques were immunized intramuscularly with CoV2 preS dTM-AS03 vaccine candidates on day 0 (D0) and on day 21 (D21). Monovalent ancestral D614 and Beta (B.1.351) vaccine candidates were used at two different antigen doses: 5 and 10 μg. Bivalent candidate D614 + Beta was used at 5 μg + 5 μg.

monovalent D614 (5 μg) on D0 and a heterologous vaccine on D21 composed of either monovalent Beta (5 μg) or bivalent D614 + Beta (5 μg + 5 μg).

Blood samples were collected before immunization on D0 and D21, as well as on D34, D70, D114, D206, and D365.

Female Golden Syrian hamsters aged of 6–8 weeks were randomized based on weight. Vaccine candidates were administrated on D0 and D21 by intramuscular route into the quadriceps of the hind leg. Twenty-four hamsters per group were vaccinated either with monovalent D614 at 1 μg or Beta at 1 μg or with bivalent D614 + Beta at 1 μg + 1 μg. Blood samples were collected before immunization (D0 and D21) and on D35. Twenty-eight days post-second dose, 8 hamsters per group were challenged by the intranasal route with NY strain (D614G) at $9.8 \times 10^3$ TCID$_{50}$ (VeroE6) or Beta at $5 \times 10^2$ TCID$_{50}$ (VeroE6) or Alpha at $1.6 \times 10^6$ TCID$_{50}$, (Vero TMPRSS2) infectious doses previously determined to result in 10–20% of body weight loss 7 days post-challenge. Body weight was monitored daily until the end of the study, i.e., 4 or 7 days post-challenge for half the number of animals per group per timepoint. Lungs were harvested 4- or 7-days post-challenge for viral load and pathology assessment.

**Convalescent human sera.** A convalescent human serum panel ($N = 93$) was obtained from commercial vendors (Sanguine Biobank, iSpecimen, and PPD). The serum samples were collected within 3 months following PCR-positive diagnosis of COVID-19 in 2020.

The WHO International Standard for anti-SARS-CoV-2 immunoglobulin (human) (NIBSC code: 20/136) was also used to allow the accurate calibration of assays to an arbitrary unit, thereby reducing inter-laboratory variation and creating a common language for reporting data.

**Pseudovirus-based virus neutralization assays.** The assay has been described previously in ref. [14]. Mu (B.1.621) and Omicron BA.4/5 variants were added to the panel of pseudoviruses (Table 1).

Serum samples were diluted 1:4 or 1:20 in media (FluoroBrite phenol red-free DMEM + 10% FBS + 10 mM HEPES + 1% PS + 1% Glutamax) and heat-inactivated at 56 °C for 30 min. Further, a twofold, 11-point, dilution series of the heat-inactivated serum were performed in media. Diluted serum samples were mixed with reporter virus particle (RVP)-GFP (Integral Molecular) listed in the table below (Table 1) diluted to contain ~300 infectious particles per well and incubated for 1 h at 37 °C. 96-well plates of ~50% confluent 293T-hsACE2 clonal cells (Integral Molecular, Cat# C-HA102) in 75 μL volume were inoculated with 50 μL of the serum + virus mixtures and incubated at 37 °C for 72 h. At the end of the 72-h incubation, plates were scanned on a

high-content imager and individual GFP-expressing cells were counted. The neutralizing antibody titer was reported as the reciprocal of the dilution that reduced the number of virus plaques in the test by 50%.

Vesicular stomatitis virus (VSV)-ancestral D614 pseudovirus qualified assay (Nexelis, Laval, Canada) was performed as described previously. Briefly, pseudotyped virus particles were produced from a modified Vesicular Stomatitis Virus (VSVΔG) backbone bearing the SARS-CoV-2 Spike (Wuhan) and expressing the luciferase reporter for detection. Twofold serial dilution of heat-inactivated sera were mixed with 75,000–300,000 relative luminescence units (RLU) and incubated at 37 °C, 5% $CO_2$ for 60 min. The mix was then transferred to 96-well plates previously seeded overnight with VeroE6 cells, and incubated at 37 °C, 5% $CO_2$ for 20 h. After the addition of the luciferase substrate, the plates were read for luminescence on a Spectramax 340PC Versamax). The intensity of luminescence quantified in RLU, is inversely proportional to the neutralizing antibodies present in the serum. Neutralizing antibody titers were calculated as the reciprocal of the serum dilution resulting in a 50% reduction of the RLU measured in the absence of serum.

**Enzyme-linked immunosorbent assay (ELISA).** S-specific IgGs were assayed using indirect ELISA. Nunc microwell plates were coated with Spike SARS-CoV S-GCN4 protein (GeneArt, expressed in Expi 293 cell line) at 0.5 μg/mL in PBS at 4 °C overnight. Plates were washed three times with PBS–Tween 0.1% before blocking with 1% BSA in PBS–Tween 0.1% for 1 h. Heat-inactivated samples were plated at a 1:450 initial dilution followed by threefold, seven-point serial dilutions in blocking buffer. Plates were washed three times after a 1-h incubation at room temperature before adding a secondary antibody. Plates were incubated at room temperature for 1 h and washed thrice. Plates were developed using Pierce 1-Step Ultra TMB-ELISA Substrate Solution for 6 min and stopped by TMB STOP solution. Plates were read at 450 nm in a SpectraMax® plate reader, and the data analyzed using Softmax® Pro 6.5.1 GxP software and the proprietary software, Sanofi Universal Exporter 2.1. Antibody titers were reported as the highest dilution that is equal to 0.2-OD cutoff.

**Enzyme-linked immunospot (ELISpot).** Memory B cells were analyzed using Human IgG Single-color B cell ELISpot kit (CTL, CAT# NC1911372). Cryo-preserved PBMCs were quickly thawed in a 37 °C water bath. A Fetal Calf Serum (FCS) /DNAse I (200 unit/mL) mixture was slowly added to PBMCs, before being transferred to a complete cell culture medium (CM) (RPMI-1640 with 10% FCS and antibiotic cocktail). After centrifugation and resuspension into 6 mL of CM, PBMCs were transferred into six-well plates and incubated at 37 °C with 5% of $CO_2$ for 1 h. Then,

B-Poly-S$^{TM}$ was added at 1:1000 dilution for cell stimulation, for 4 days at 37 °C with 5% of $CO_2$.

Pre-stimulated PBMCs were harvested and centrifuged at $433 \times g$ for 5 min at room temperature (RT). After washing, PBMCs were counted using Guava® easyCyte cell counter and the cells were adjusted to the desired concentration with CM.

Ninety-six wells-plates with PVDF membrane were permeabilized with 15 μL of 70% ethanol for a maximum of 1 min, then washed three times with sterile phosphate-buffered saline (PBS) 1X before being coated with 80 μL of human Ig capture antibody (Ab) diluted at 1:50 or with SARS-CoV2 S-GCN4 protein, Wuhan strain (GeneArt, expressed in Expi 293 cell line) or BA.5 strain (Sino Biologicals, expressed in HEK293 cells) at 4 μg/mL or with PBS. The plates were incubated overnight at 4 °C and then washed three times with sterile PBS and with CM for 1 h at RT. CM was then removed and PBMCs were added at $3 \times 10^5$ cells/well in the S-GCN4 protein (Wuhan or BA.5 strain) and PBS-coated wells, and at $5 \times 10^3$ cells in the Ig capture Ab-coated wells, under 100 μL/well. Each condition was tested in duplicate, and plates were incubated for 18 h at 37 °C with 5% $CO_2$.

To reveal antibody-secreting cells, plates were first washed twice with PBS, then twice with 0.05% Tween PBS and then anti-human IgG detection solution was added under 80 μL. After 2 h of incubation at RT, plates were washed three times with 0.05% Tween PBS and a tertiary solution was added under 80 μL. Plates were incubated for 1 h at RT, then washed twice with 0.05% Tween PBS and twice with distilled water. A blue developer solution was added under 80 μL and incubated at RT for 15–20 min. The reaction was stopped by rinsing the plate membrane with water and decant three times. The plates were air-dried, then scanned and read using Cytation 7 analyzer. The number of spots with the PBS only (background) was subtracted from the number of S-specific or total IgG spots. The results are expressed as S-specific IgG-secreting memory B cells/million PBMCs and as % of S-specific IgG-secreting memory B cells among all circulating IgG-secreting memory B cells.

**Antibody-dependent cellular phagocytosis (ADCP).** THP-1 cells (ATCC) were maintained in RPMI-1640 (Sigma-Aldrich) supplemented with 10% fetal bovine serum (FBS), 5% penicillin/streptomycin (Corning, 50 μg/mL), 5% L-glutamine (Corning, 4 mM), 5% HEPES buffer (pH 7.2) (Corning, 50 mM), and 0.5% 2-Mercaptoethanol (Gibco, 275 μM) at 37 °C, 5% $CO_2$. ADCP was performed as previously described in ref. [23]. Briefly, D614 Spike, Beta Spike, or Omicron Spike (Sino Biological) were biotinylated and coupled to yellow-green NeutrAvidin FluoSpheres (Invitrogen). Immune complexes were formed by mixing antigen-coupled beads with serum diluted 1:100 in PBS. Immune complexes were incubated for 2 h at 37 °C and washed in PBS. THP-1 cells were added to immune complexes at a concentration of $1.25 \times 10^5$ cells/mL and incubated overnight at 37 °C, 5% $CO_2$. The ability of antibodies to drive bead uptake by THP-1 cells was assessed by flow cytometry using a BD LSR II cytometer. PhagoScores were calculated as follows: (% bead + cells × GeoMean of cells)/10000. Samples were run in duplicate, and the data represent the average of the duplicates.

**Antibody-dependent neutrophil phagocytosis (ADNP).** ADNP was performed as previously described in ref. [24]. Peripheral whole blood was collected by the Ragon Institute from healthy volunteers. Volunteers provided signed consent, were over 18 years old, and were deidentified. The study was approved by the MGH Institutional Review Board. Red blood cells were lysed by ammonium chloride potassium (ACK) lysis. White blood cells were washed with PBS and maintained in RPMI-1640 (Sigma-

Aldrich) media supplemented with 10% fetal bovine serum (FBS) (Sigma-Aldrich), 5% penicillin/streptomycin (Corning, 50 μg/mL), 5% L-glutamine (Corning, 4 mM), 5% HEPES buffer pH 7.2 (Corning, 50 mM) and 37 °C, 5% $CO_2$ for the duration of the assay. Yellow-green NeutrAvidin FluoSpheres were coupled to antigen as described for ADCP. Immune complexes were formed by mixing serum diluted 1:50 in PBS with coupled beads and incubating for 2 h at 37 °C. Immune complexes were washed and white blood cells were added at a concentration of $2.5 \times 10^5$ cells/mL. Cells were incubated with immune complexes for 1 h at 37 °C. PacBlue anti-CD66b (BioLegend, clone: UCH71) was used to stain for neutrophils. Phagocytosis was measured by flow cytometry using an iQue (Intellicyt). Phagocytosis by neutrophils (CD66b+) was calculated as described for ADCP. The experiment was performed with two donors and the reported value is the average of the two donors.

**Antibody-dependent complement deposition (ADCD).** ADCD was performed as previously described in ref. [25]. Red NeutrAvidin FluoSpheres were coupled to antigen as described for ADCP. Immune complexes were formed by mixing serum diluted 1:10 in PBS with coupled beads and incubating for 2 h at 37 °C. Immune complexes were washed and lyophilized guinea pig complement (Cedarlane) diluted in gelatin veronal buffer with calcium and magnesium (Sigma-Aldrich) was added to immune complexes and incubated for 20 min at 37 °C. C3 deposition was measured by anti-guinea pig C3 FITC (MpBio). Fluorescence was acquired using a BD LSR II cytometer. C3 deposition is reported as the median fluorescence intensity of FITC. The experiment was performed in duplicate, and the reported value is the average of the two replicates.

**Antibody-dependent natural killer cell (NK) activation (ADNKA).** ADNKA was performed as described previously in ref. [26]. ELISA plates were coated with 2 μg/mL of antigen and incubated for 2 h at 37 °C. Plates were washed with PBS and blocked overnight with 5% bovine serum albumin (BSA) at 4 °C. Buffy coats were collected by Massachusetts General Hospital from healthy donors who were over 18 years old and provided signed consent. Samples were deidentified before use. NK cells were isolated from buffy coats using RosetteSep (STEMCELL Technologies) and then separated using a ficoll gradient. NK cells were rested overnight at 37 °C, 5% $CO_2$ in R10 (RPMI-1640 (Sigma-Aldrich) media supplemented with 10% fetal bovine serum (FBS) (Sigma-Aldrich), 5% penicillin/streptomycin (Corning, 50 μg/mL), 5% L-glutamine (Corning, 4 mM), 5% HEPES buffer (pH 7.2) (Corning, 50 mM)) supplemented with 2 ng/mL IL-15. The following day, plates were washed, and samples diluted 1:25 in PBS were added to the plates. Plates were incubated for 2 h at 37 °C, washed, and NK cells were added at a concentration of $2.5 \times 0^5$ cells/mL in R10 media supplemented with anti-CD107a–phycoerythrin (PE)–Cy5 (BD Biosciences, lot # 0149826, 1:1000 dilution), brefeldin A (10 μg/ml) (Sigma-Aldrich), and GolgiStop (BD Biosciences). Plates were incubated for 5 h at 37 °C. Following the incubation, cells were stained for surface markers with anti-CD3 Pacific Blue (BD Biosciences, clone G10F5), anti-CD16 allophycocyanin (APC)-Cy5 (BD Biosciences, clone 3G8), and anti-CD56 PE-Cy7 (BD Biosciences, clone B159) for 15 min at room temperature. Cells were fixed with PermA (Life Technologies) and permeabilized with PermB (Life Technologies) and stained with anti-MIP-1β PE (BD Biosciences) and anti-IFN gamma FITC for 15 min at room temperature. Fluorescence was analyzed by flow cytometry using a BD LSR II. NK cells were gated as CD56 + CD16 + CD3− and activity was determined as the percent of NK cells positive for

CD107a, MIP-1b, or IFNg. The assay was performed with two donors and the data reported represents the average of the two donors.

**Luminex**. Antigen-specific antibody isotype titer and Fc receptor (FcR)—the binding was determined by a multiplex Luminex assay, as previously described in ref. [27]. Carboxylated megaplex microspheres (Luminex) were covalently linked to antigens using NHS-ester linkages by the addition of Sulfo-NHS and EDC (Thermo Fisher). Immune complexes were formed by adding a diluted serum to antigen-coupled microspheres, and plates were incubated overnight at 4 °C, shaking at 700 rpm. The following day, plates were washed with 0.1% BSA and 0.02% Tween-20. For the detection of antibody isotype titer, PE-coupled mouse anti-human detection antibodies (Southern Biotech) were added to the plates. For the detection of FcR-binding, Avi-tagged human FcRs (Duke Human Vaccine Institute) were biotinylated using a BirA500 kit (Avidity) per manufacturer's instructions and tagged with streptavidin-PE. The PE-tagged FcR was added to the immune complex. Fluorescence was acquired using an iQue (Intellicyt), and the data represent the median fluorescence intensity (MFI). The Luminex assay was run in duplicate, and the data reported represents the average of the duplicates. Each Luminex assay was performed with multiple dilutions for the same antigen. Representative data were shown.

**TCID$_{50}$ assay**. The TCID$_{50}$ assay was conducted by the addition of tenfold graded dilutions of samples to TMPRSS2 monolayers. Specifically, Vero TMPRSS2 cells (obtained from Adrian Creanga, Vaccine Research Center-NIAID) were plated at 25,000 cells/well in DMEM + 10% FBS + Gentamicin and the cultures were incubated at 37 °C, 5% $CO_2$. The medium was aspirated and replaced with 180 µL of DMEM + 2% FBS + gentamicin. Twenty (20) µL of the sample was added to the top row in quadruplicate and mixed using a P200 pipettor five times. Using the pipettor, 20 µL was transferred to the next row, and repeated down the plate (columns A–H), representing tenfold dilutions. The tips were disposed for each row and repeated until the last row. Positive (virus stock of known infectious titer in the assay) and negative (medium only) control wells were included in each assay set-up. The plates were incubated at 37 °C, 5% $CO_2$ for 4 days. The cell monolayers were visually inspected for cytopathic effect (CPE). The TCID$_{50}$ value was calculated using the Read-Muench formula. For samples which had less than 3 CPE positive wells, the TCID$_{50}$ could not be calculated using the Reed-Muench formula, and these samples were assigned a titer below the limit of detection (i.e., challenge prototype D614G 1.569 TCID$_{50}$/gram, challenge Alpha 1.652 TCID$_{50}$/gram and challenge Beta 1.550 TCID$_{50}$/gram). For optimal assay performance, the TCID$_{50}$ value of the positive control should test within twofold of the expected value.

**Histopathology**. Whole left hamster lung samples were collected in 10% neutral buffered formalin (NBF), routinely processed, and paraffin-embedded. Paraffin blocks were sectioned at approximately 5 microns and stained using hematoxylin and eosin (H&E). A board-certified veterinary pathologist blindly evaluated H&E slides both qualitatively and semi-quantitatively for pulmonary pathology. Semi-quantitative scores were made based on the percent of lungs affected. Lungs that were consistent with normal tissue were given a score of 0. If less than 25% of the section revealed histopathology, a score of 1 was assigned. A score of 2 was attributed to lung sections in which histopathology was present in greater than 25% but less than 50% of the lung. If more than 50% of the lung parenchyma was involved, a score of 3 was

assigned. Nucleocapsid protein expression was assessed by DAB chromogen-based immunohistochemistry with a rabbit polyclonal anti-SARS-CoV-2 nucleocapsid antibody (GTX135357, GeneTex, Inc, USA) in Leica BondRX automated immunostaining platform (Leica, USA) as previously described[28]. Nucleocapsid immuno-positive area was measured in lung sections using a percent area algorithm of HALO image analysis software (Indica Labs, USA).

**Statistics and reproducibility**. The power of the study was estimated based on the pseudovirus-neutralizing titer, considering that the minimal pertinent difference corresponded to threefold (0.5 log), with a standard deviation of 0.3 log (i.e., observed in previous studies). In such context, a sample size of 6 NHP per group was considered acceptable to show a difference between the 11 groups with a power upper to 75%.

At the time of the assignment, the characteristics at baseline (sex, age, and weight) were balanced to have comparable groups. ELISA titers and neutralizing titers were $\log_{10}$ transformed prior to statistical analysis.

All statistical tests were two-sided, the nominal level of statistical significance was set to $\alpha = 0.05$ for effect size estimates, $\alpha = 0.01$ for normality tests and $\alpha = 0.10$ for interaction terms. The analyses were performed on SEG SAS v9.4®.

For all the readouts of NHP studies and the ELISA titers of the hamster study, analyses were performed using mixed models with product, time, and their interaction as fixed factors, a time considered as repeated, and cohorts added as a random effect for the hamster study. For neutralizing titers of the hamster study as well as Omicron BA.5 Spike-specific IgG-secreting memory B cell responses of the NHP study, as only one timepoint data were available, a one-way ANOVA using a mixed model with a product as a fixed factor and cohorts as a random effect was performed. The normality of the residual of the model were checked using a Normal Probability Plot, and the results were considered acceptable.

Spearman correlation between PsV titers and weight loss were performed.

For system serology univariate analysis, statistics were calculated using GraphPad Prism version 8.0. Significance was determined by a Kruskal–Wallis test. Heatmaps and polar plots were created in Python (version 3.9.1). The polar plots show the median percentile rank for each feature. The heatmaps show the $z$-scored data.

**Reporting summary**. Further information on research design is available in the Nature Research Reporting Summary linked to this article.

## Results

**Ancestral/Beta bivalent CoV2 preS dTM-AS03 confers more balanced and broader neutralizing antibody responses than D614 or Beta monovalent vaccines in naïve NHPs.** We first checked for potential immune interference between the two valences in the bivalent formulation as compared to the D614 or Beta monovalent formulations. Naïve cynomolgus macaques were immunized on D0 and D21 with the bivalent CoV2 preS dTM-AS03 (D614 + Beta) vaccine formulation at 5 µg + 5 µg dose, or monovalent CoV2 preS dTM-AS03 vaccines (D614 or Beta) at 5 µg and 10 µg doses (Fig. 1).

Neutralizing antibody responses against the ancestral Wuhan strain D614 and the Beta VOC were analyzed two weeks post-dose 2 using lentivirus-based pseudovirus assays (lentivirus-PsV). The bivalent vaccine candidate elicited high and consistent nAb titers against ancestral SARS-CoV-2 D614, with mean titers of 3.0

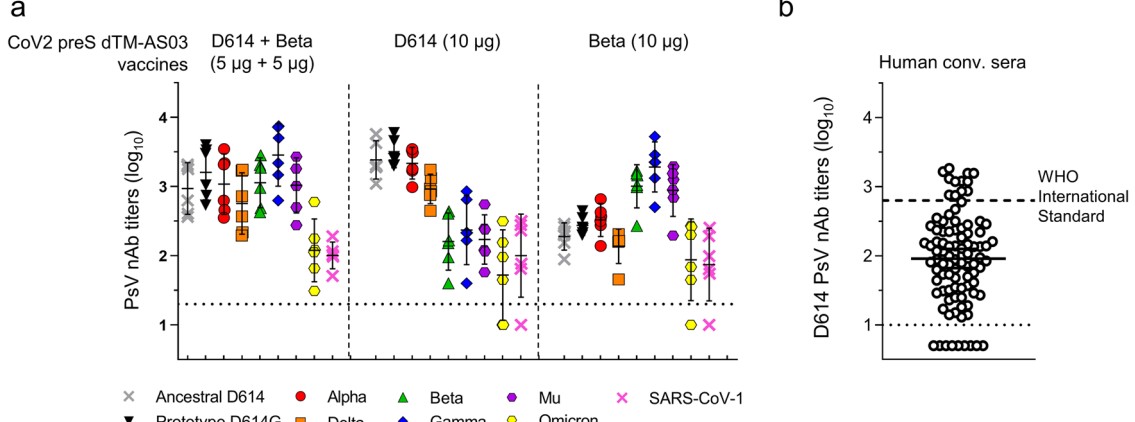

**Fig. 2 Bivalent CoV2 preS dTM-AS03 (D614 + Beta) induces broader neutralization of variants as compared to monovalent vaccines.** Groups of six naïve macaques were immunized twice at 3 weeks interval with the bivalent vaccine (5 μg + 5 μg), monovalent D614 or monovalent Beta (10 μg). Neutralizing antibody titers were measured 2 weeks post-second dose. **a** Lentivirus-based pseudovirus-neutralizing antibody titers against the ancestral D614 and prototype D614G, variants of concern (Alpha, Beta, Gamma, Delta, Mu, and Omicron BA.1) and SARS-CoV-1 at 2 weeks post-dose 2. Individual macaque data were shown. **b** A human convalescent panel of sera (N = 93) was assayed against D614 ancestral pseudovirus and is plotted as a comparator. The dashed line represents the World Health Organization (WHO) International Standard for anti-SARS-CoV2 immunoglobulin (human) NIBSC code: 20/136. Horizontal dotted lines indicate the limits of quantification of the assay. Bars represent means and 95% confidence intervals.

$\log_{10} \pm 0.4 \log_{10}$ standard deviation (SD) (Supplementary Fig. 1a), not significantly different from those induced by the monovalent D614 at 5 μg (mean titers of 3.1 $\log_{10}$) and 10 μg (mean titers of 3.4 $\log_{10}$), but significantly higher than those elicited by the monovalent Beta vaccine at 10 μg (mean titers of 2.3 $\log_{10}$, $P = 0.0017$, mean fold increase 4.9). The absence of negative immune interferences in the bivalent formulation was confirmed using a qualified D614 vesicular stomatitis virus (VSV)-pseudovirus assay (Supplementary Fig. 1b).

As expected, the bivalent vaccine also elicited high and consistent nAb titers against Beta pseudovirus in all macaques, with a mean of 3.0 $\log_{10} \pm 0.3 \log_{10}$ SD, equivalent to those induced by the monovalent Beta vaccine at 5 μg and 10 μg (Supplementary Fig. 1c) and significantly higher than those induced by the monovalent D614 at 10 μg (sevenfold mean increase, $P < 0.001$).

Overall, the bivalent CoV2 preS dTM-AS03 induced robust and balanced nAb titers against both D614 ancestral strain and Beta variant, with no detectable immune interferences.

We next assessed the breadth of neutralization conferred by the bivalent vaccine against other VOC, D614G, Alpha, and Delta (all containing the original E484 position), Gamma and Mu (containing the E484K mutation present in Beta) and the more distant Omicron BA.1 (containing E484A mutation among 15 mutations in the RBD) and SARS-CoV-1. The bivalent vaccine was compared to the monovalent vaccine formulations at the same timepoint (i.e., 2 weeks post-second dose) (Fig. 2a).

The bivalent vaccine elicited high and homogenous nAb titers against all VOC—except Omicron BA.1—with mean titers ranging from 2.8 $\log_{10}$ against Delta to 3.5 $\log_{10}$ against Gamma. When compared to the monovalent D614 vaccine (10 μg), the bivalent vaccine elicited equivalent nAb titers against D614, D614G, Alpha, and Delta (mean titers of 3 to 3.5 $\log_{10}$), but higher nAb titers against Beta, Gamma, and Mu (mean titers of 2.2 to 2.4 $\log_{10}$, $P < 0.001$, mean fold-increase from 6.1 to 12.2). Conversely, compared to the Beta monovalent, the bivalent vaccine induced higher nAb titers against D614, D614G, Alpha, and Delta (mean titers of 2.1 to 2.5 $\log_{10}$, mean fold increase from 3.3 to 5.8, $P < 0.05$) but equivalent nAb titers against Beta, Gamma, and Mu (mean titers of 2.9 to 3.3 $\log_{10}$).

For Omicron BA.1 and SARS-CoV-1, the bivalent vaccine induced detectable titers in all animals, with mean titers around 2.0 $\log_{10}$, whereas nAb responses were more variable in the monovalent vaccine groups.

The results indicate that the ancestral/Beta bivalent vaccine candidate elicits robust and balanced nAb responses against the original E484-containing viruses (D614, D614G, Alpha, and Delta), the E484K-containing viruses (Beta, Gamma, and Mu) and lower but consistent nAb responses against the distant Omicron BA.1 (E484A) and SARS-CoV-1.

For comparison, the bivalent vaccine elicited mean D614 nAb titers higher than that measured in the World Health Organization (WHO) International Standard for anti-SARS-CoV-2 immunoglobulin (NIBSC code: 20/136; 2.8 $\log_{10}$ in our lentivirus-pseudovirus assay) and higher than those measured in a panel of 93 human convalescent sera (collected within 3 months after positive PCR test; 2.0 $\log_{10}$; Fig. 2b). The International Standard for anti-SARS-CoV-2 immunoglobulins, adopted by the WHO Expert Committee on Biological Standardization on Dec 10, 2020 allows the comparison of the titers against D614 generated in different neutralization assays, it has been assigned potency of 1000 UI/mL[29].

**Two-dose primary series with the ancestral/Beta bivalent CoV2 preS dTM-AS03 vaccine induces more balanced and more homogeneous nAb titers than a heterologous prime/boost regimen in naïve NHPs.** To assess the benefit of heterologous prime/boost immunization on cross-neutralization, two alternative immunization regimens were tested, where a first dose with the monovalent D614 (5 μg) (D0) was followed by a second dose (D21) with either the monovalent Beta (5 μg) or the bivalent (5 μg + 5 μg) vaccines (Supplementary Fig. 2a).

Compared to the monovalent D614 vaccine 2-dose regimen, the second injection with monovalent Beta was able to extend the breadth of the neutralizing response to Beta and other VOC, while the second injection with the bivalent didn't improve the breadth. However, the heterologous prime/boost with monovalent Beta for the second injection induced higher variability and slightly lower titers than the bivalent vaccine 2-dose regimen (not

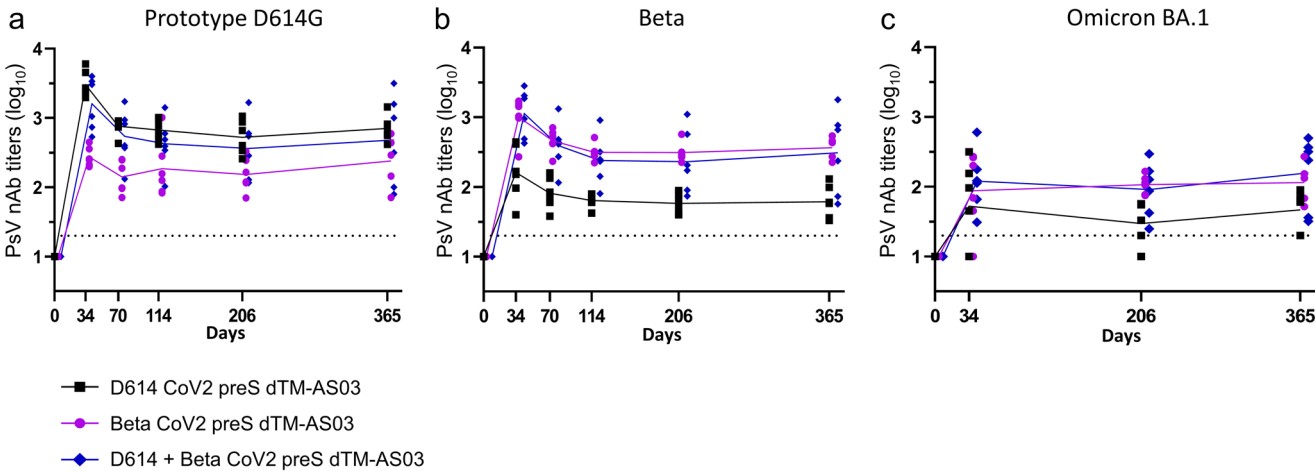

**Fig. 3 The bivalent CoV2 preS dTM-AS03 (D614 + Beta) vaccine elicits durable (1 year) neutralizing antibody titers in NHPs.** Pseudovirus-neutralizing antibody titers against **a** the prototype D614G SARS-CoV-2, **b** the Beta variant, and **c** the Omicron variant (BA.1) were assessed at different timepoints after immunization in cynomolgus macaques with monovalent D614 (10 μg), monovalent Beta (10 μg) or bivalent D614 + Beta (5 μg + 5 μg). Individual macaque data are shown ($N = 6$/group). Connecting lines indicate mean responses and the horizontal dotted line is the limit of quantification of the assay.

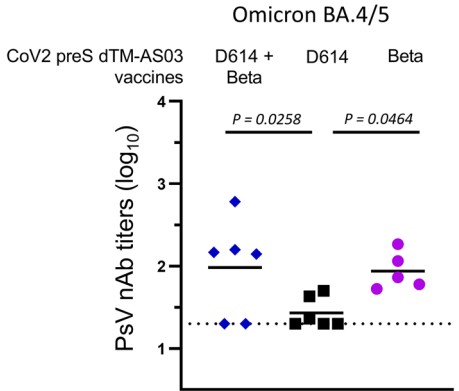

**Fig. 4 The bivalent vaccine D614 + Beta elicits cross-neutralization of the variant Omicron BA.4/5 at 1 year.** Pseudovirus-neutralizing antibody titers against Omicron BA.4/5 in individual macaque sera are shown ($N = 6$/group; one sample was missing in the Beta monovalent vaccine group). Blue diamonds: bivalent D614 + Beta formulation; black squares: D614 formulation; purple dots: Beta formulation. Bars indicate mean responses and horizontal dotted lines correspond to the inverse of the lowest dilution of the assay.

significant for D614, D614G, Alpha, and Delta mean nAb titers, 3- and 4.1-fold lower for Beta, $P = 0.0165$ and Gamma, $P = 0.0052$ mean nAb titers, respectively) (Supplementary Fig. 2b).

**The ancestral/Beta bivalent CoV2 preS dTM-AS03 vaccine elicits durable neutralizing antibody titers in naïve NHPs.** To address potential neutralizing Ab waning, we measured the nAb titers against the prototype D614G, Beta, and Omicron BA.1 in the blood of vaccinated animals up to 1-year post-immunization (D365) with the bivalent and monovalent vaccine formulations at 10 μg total antigen dose (Fig. 3).

For each vaccine formulations, nAb titers against the prototype D614G and the Beta variant peaked on D34, decreased through D70 or D114, and then remained stable for up to 1 year (D365). At 1 year, the mean nAb titers against the prototype D614G and the Beta variant were, respectively, 2.7 and 2.5 $\log_{10}$ for the

bivalent vaccine, 2.8 and 1.8 $\log_{10}$ for the D614 monovalent, and 2.4 and 2.6 $\log_{10}$ for the Beta monovalent (Fig. 3a, b).

Neutralizing titers against Omicron BA.1 were assessed on D0, D34, D206, and D365 and against BA.4/5 at 1 year. All animals had detectable BA.1 PsV nAb titers up to one year, except in the monovalent D614 vaccine group (Fig. 3c). The mean nAb titers against Omicron BA.1 (Fig. 3c) and BA.4/5 (Fig. 4) at 1 year were, respectively, 2.2 and 2.0 $\log_{10}$ for the bivalent, 1.7 and 1.7 $\log_{10}$ for the monovalent D614 and 2.1 and 1.7 $\log_{10}$ for the monovalent Beta.

The Alpha and Gamma mean nAb titers at 3 months (D114) were, respectively, 2.6 and 2.6 $\log_{10}$ for the bivalent vaccine groups, 2.7 and 2.1 $\log_{10}$ for the monovalent D614, 2.3 and 2.8 $\log_{10}$ for the monovalent Beta (Supplementary Fig. 3a, b). The Delta variant mean nAb titers at 7 months (D206) were, 2.1 $\log_{10}$ (bivalent), 2.2 $\log_{10}$ (monovalent D614), and 1.8 $\log_{10}$ (monovalent Beta) (Supplementary Fig. 3c).

Interestingly, the decay appeared less pronounced for cross-neutralizing responses than for vaccine-homologous neutralizing responses. Of note, the cross-neutralizing responses tended to peak at lower titers than vaccine-homologous neutralizing responses (Fig. 3 and Supplementary Fig. 3). Thus, the difference between homologous and heterologous neutralizing titers were less at 1-year post-immunization than at 2 weeks post-second dose. This holds especially for Omicron BA.1 for which no significant nAb titers decline was observed with all three vaccine formulations between D34 and 1 year.

Modeling the Ab decay based on the experimental data indicated that, after the initial decay, prototype D614G nAb titers reached a plateau in the bivalent group on D98 at 2.6 $\log_{10}$ and stayed stable up to 1 year (D365). For the monovalent D614 10 μg group, a plateau was reached on D89, with the mean titer estimated at 2.8 $\log_{10}$. In the monovalent Beta 10 μg group, the plateau was reached on D34 at 2.3 log10, indicating no D614G nAb decay over one year in this group.

Beta nAb titers reached a plateau for all groups, estimated at 2.4 $\log_{10}$ on D111, 1.8 $\log_{10}$ on D113, 2.5 $\log_{10}$ on D114 for bivalent, monovalent D614, and monovalent Beta, respectively.

Regarding Omicron BA.1 nAb titers, no mathematical model was performed due to the few timepoints (D34, 206, and 365) assessed post-immunization, nevertheless, no statistically significant time effect was observed from D34 to 1 year (D365), meaning the neutralizing titers were stable over 1 year.

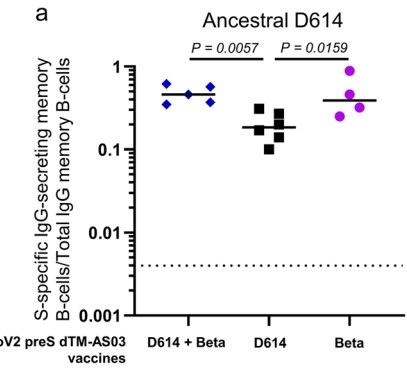 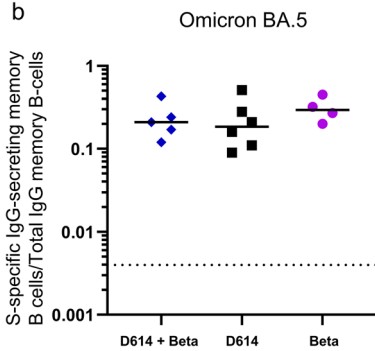

**Fig. 5 CoV2 preS dTM-AS03 bivalent (D614 + Beta) vaccine induces memory B cells at 1 year specific of the ancestral D614 and BA.5 spikes in NHPs.**
Spike-specific IgG-secreting memory B cell responses at 1 year induced by the bivalent and monovalent CoV2 preS dTM-AS03 vaccines against **a** ancestral D614 spike and **b** Omicron BA.5 spike. Symbols represent the individual frequency of S-specific/total IgG-secreting memory B cells (%) in macaques immunized with the bivalent vaccine (5 µg + 5 µg) (blue diamonds), monovalent D614 (10 µg) (black squares) or monovalent Beta (10 µg) (purple dots) and bars the median of each group.

**The ancestral/Beta bivalent CoV2 preS dTM-AS03 vaccine elicits long-term memory B cell responses in naïve NHPs.** At 1 year, ancestral D614 and Omicron BA.5 Spike-specific IgG-secreting memory B cell responses were detected for all animals at similar levels across all three vaccine groups, i.e., bivalent, monovalent D614 and monovalent Beta (Fig. 5). When adjusted to total IgG memory B cells, the responses were very homogeneous within groups. The medians of ancestral D614 S-specific/total IgG-secreting memory B cells were 0.46, 0.19, and 0.39% for the bivalent, monovalent D614, and monovalent Beta vaccine groups, respectively. Significantly higher ancestral D614 Spike-memory B cells were detected in the bivalent and monovalent Beta groups compared to the monovalent D614 formulation. The medians of Omicron BA.5 S-specific/total IgG-secreting memory B cells were 0.21, 0.19, and 0.29% for the bivalent, monovalent D614 and monovalent Beta vaccine groups, respectively, with no significant differences between groups.

**Ancestral/Beta bivalent CoV2 preS dTM-AS03 vaccine induces similar antibody Fc functions as monovalent vaccines.** In addition to neutralizing antibodies, Fc receptor-binding and antibody effector functions were assessed as they have been previously shown to contribute to the clearance of SARS-CoV-2 infection[30]. The bivalent formulation elicited similar IgG1 titer and FcR2a-binding antibodies against D614 and Beta Spikes compared to the monovalent vaccine formulations at 5 and 10 µg (Fig. 6a, b).

We then measured the ability of the antibodies from these different vaccine formulations to induce cellular effector functions. We found that the bivalent formulation induced significantly less antibody-dependent cellular phagocytosis (ADCP) against D614 Spike compared to the monovalent formulation at 10 µg (Fig. 6c). No significant differences were observed for the ability of the antibodies to elicit antibody-dependent complement deposition (ADCD) (Fig. 6d).

To understand the difference between the bivalent and monovalent formulations on a broad level, we plotted a heatmap of the z-score of all measured antibody titers, FcR-binding, and antibody effector functions against D614 (Fig. 6e, left panels) or Beta (Fig. 6e, right panels). This analysis did not reveal any specific pattern that discriminates the bivalent from the monovalent formulations. These data highlight that the bivalent formulation induces robust IgG subclass antibody titer and FcR-binding function, comparable to the monovalent formulations.

Further analyses of the effector functions induced by the different formulations were performed against the Omicron

variant (BA.1) Spike. The IgG1 binding titer to the Omicron Spike was highly correlated to the IgG1 D614G binding (Supplementary Fig. 4a), but showed a decreased binding as shown in previous studies[31]. The bivalent formulation induced Spike-specific antibodies against Omicron with similar titers (Supplementary. Fig. 4b) and similar FcR2a and FcR3a binding (Supplementary Fig. 4c) compared to the monovalent formulations D614 and Beta at 5 and 10 µg. Whereas all formulations induced similar antibody-dependent cellular and neutrophil phagocytosis (ADCP and ADNP, respectively), the 10 µg Beta monovalent formulation induced higher ADCD than the 5 µg Beta monovalent formulation, and trended higher than all other formulations (Supplementary Fig. 4d). Whether the increase in ADCD activity is simply due to higher vaccine dose or the induction of IgM against Omicron is unclear.

Finally, to understand whether the bivalent vaccine formulation induces a wider breadth of antibody response, we plotted the median percentile rank of the IgG1 titers for the bivalent formulation and the four monovalent formulations against D614G, Omicron, Alpha, Beta, and Delta Spike (Supplementary Fig. 4e). This analysis suggests that the bivalent formulation and the two monovalent D614 and Beta formulations at 10 µg may offer a broader IgG response than the monovalent formulations at 5 µg.

**The ancestral/Beta bivalent CoV2 preS dTM-AS03 vaccine formulation induces robust and balanced neutralizing antibody responses against ancestral SARS-CoV-2 and VOC in hamsters.** We next investigated the protection conferred by the bivalent vaccine in Golden Syrian Hamsters against viral replication and lung pathology induced by viral challenge with the prototype D614G, Alpha and Beta variants. Three cohorts of 32 hamsters (each divided into four groups of eight animals) were immunized with either, buffer (control group), monovalent D614, monovalent Beta, or bivalent (D614 + Beta) CoV2 preS dTM-AS03, using the two-dose D0/D21 regimen, by intramuscular route at 1 µg of each antigen (Fig. 7a). Four weeks post-second dose, each cohort was challenged with either, the prototype D614G, Alpha or Beta viruses, using infectious doses previously characterized to induce between 10 and 20% body weight loss.

Before the challenge, S-specific IgG (ELISA) and nAb titers against prototype D614G, Alpha, Beta, and Delta VOC were evaluated in sera from vaccinated and non-vaccinated hamsters, using lentivirus-pseudovirus assays. The different vaccine formulations induced S-specific IgGs in all hamsters 21 days after the first injection except for 1 and 2 hamsters in the D614 and

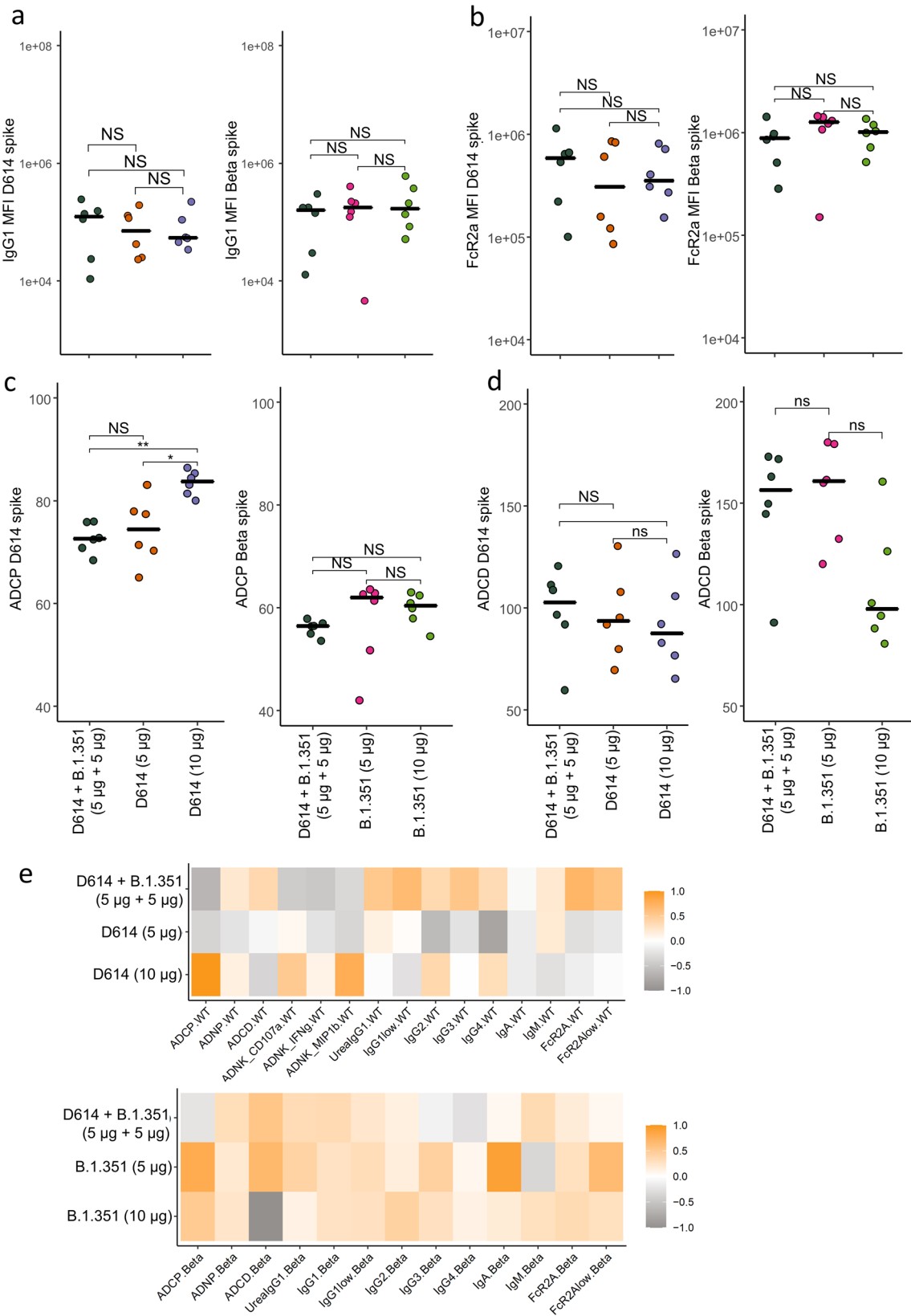

Beta monovalent vaccine groups, respectively. Mean IgG titers varied from 3.2 to 3.5 $\log_{10}$ ELISA Units (EU), as shown in Fig. 7b, and were comparable in all the vaccine groups. The mean S-specific IgG titers increased to ~5.0 $\log_{10}$ EU, 2 weeks post-second dose, with no statistically significant differences between the different vaccine formulations. Consistent with the prior

observations in macaques, the bivalent vaccine formulation elicited high and balanced nAb titers at D35 against prototype D614G, Alpha, Delta, and Beta with mean titers between 2.7 to 3.2 $\log_{10}$ (Fig. 7c). The monovalent D614 formulation induced high and balanced nAb titers against prototype D614G, Alpha, and Delta (mean titers between 2.9 and 3.1 $\log_{10}$), whereas nAbs

**Fig. 6 CoV2 preS dTM-AS03 Bivalent (D614 + Beta) vaccine induces similar antibody Fc functions as monovalent vaccines in NHPs. a** The dot plots show the IgG1 titer against D614 (left) or Beta (right) Spike. **b** The dot plots show the FcR2a-binding titer against ancestral D614 (left panels) or Beta (right panels) Spike. **c** The dot plots show the antibody-dependent cellular phagocytosis (ADCP) against ancestral D614 (left) or Beta (right) Spike. **d** The dot plots show the antibody-dependent complement deposition (ADCD) against ancestral D614 (left) or Beta (right) Spike. **e** The heatmaps show the median z-score for all antibody features measured against ancestral D614 Spike (left) or Beta Spike (right). Significance was determined by a Kruskal–Wallis test followed by post hoc Benjamini–Hochberg $p$ value correction for multiple comparisons. MFI mean fluorescence intensity. **a–d** Symbols represent individual values in macaques immunized with the bivalent vaccine (5 µg + 5 µg) (black dots), monovalent vaccines at 5 µg (red dots), or monovalent vaccines at 10 µg (green dots) and bars the median of each group.

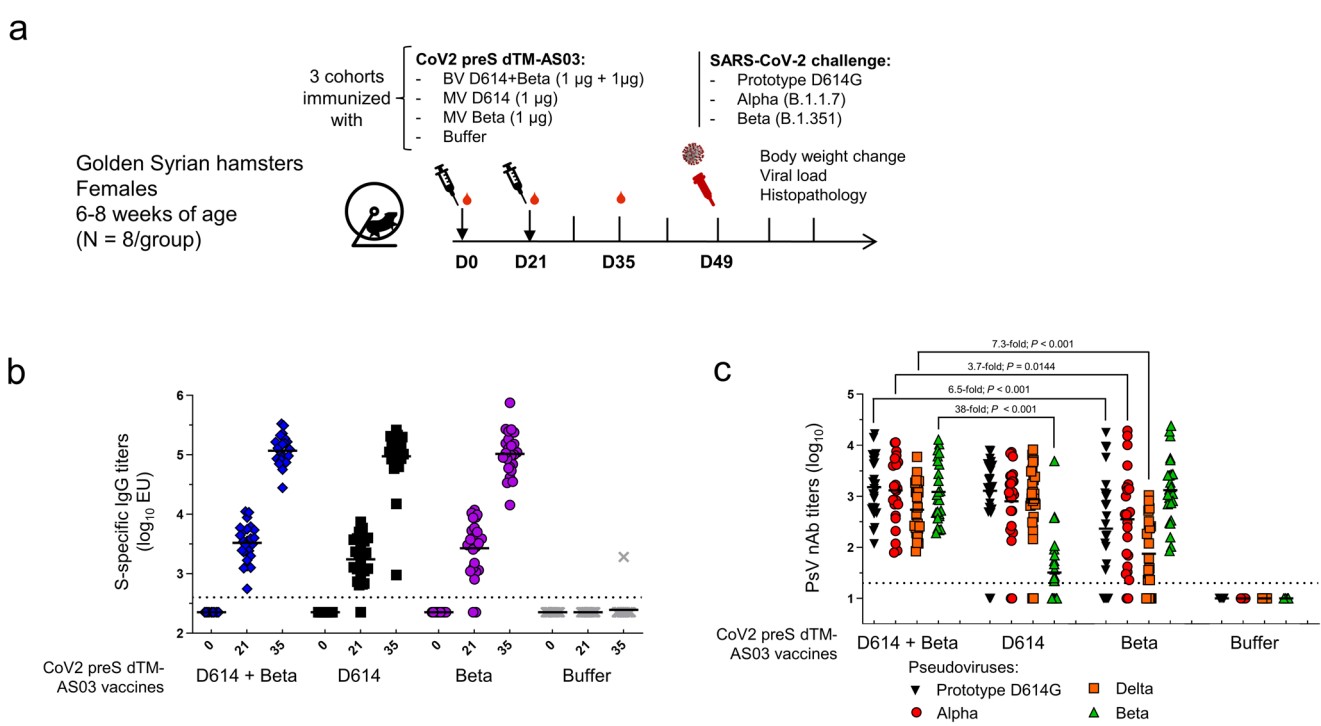

**Fig. 7 Antibody responses elicited by the bivalent and monovalent CoV2 preS dTM-AS03 vaccines in hamsters. a** Study schema. Hamsters were vaccinated with the bivalent D614 + Beta (1 µg + 1 µg) and the monovalent D614 or Beta (1 µg), on D0 and D21. **b** S-specific IgG titers were assayed in individual hamsters immunized with the bivalent vaccine (blue diamonds), the D614 monovalent vaccine (black squares), or the Beta monovalent vaccine (purple dots) on D0, D21 (after the first dose), and D35 (after second dose). **c** PsV nAb titers against prototype D614G, Alpha, Delta, and Beta variants were assayed on D35 in vaccinated and non-vaccinated hamsters. Symbols represent individual data and bars the mean of the group. Horizontal dotted lines correspond to the inverse of the lowest dilution of the assays.

were lower against Beta (mean titers of 1.5 log$_{10}$). It is noteworthy that two hamsters vaccinated with the monovalent D614 vaccine had low S-specific IgG titers (3.0 and 4.2 log$_{10}$ EU) and no detectable nAb titers, 2 weeks post-second dose. Conversely, the monovalent Beta formulation induced only high nAb titers against the homologous Beta pseudovirus (mean titers of 3.1 log$_{10}$) and lower nAb titers against prototype D614G, Alpha, and Delta (mean titers between 1.9 and 2.5 log$_{10}$; Fig. 7c). Those data are in line with our observations in macaques showing no immune interferences between the two antigens.

Statistically significant higher nAb titers against Beta were observed with the bivalent formulation compared to the monovalent D614 formulation (38-fold, $P < 0.001$), and for the prototype D614G (6.5-fold, $P < 0.001$), Alpha (3.7-fold, $P = 0.0144$) and Delta (7.3-fold, $P < 0.001$) nAb titers compared to the monovalent Beta vaccine.

**The ancestral/Beta bivalent CoV2 preS dTM-AS03 vaccine confers protection against D614G, Alpha and Beta variants replication and pathology in hamsters.** To assess protection against prototype (D614G) and variants (Alpha and Beta), body

weight change was monitored up to 7 days post-challenge as a marker of disease progression, and lung viral loads and pathology were assessed after necropsy on D4 or D7 on half the animals. Unvaccinated hamsters (buffer) lost up to 18% of body weight 7 days post-challenge with Beta and Alpha and up to 10% after prototype D614G challenge (Fig. 8a), while non-challenged hamster weights remained stable or slightly increased during the same period. Except for the two low responder hamsters (with no nAb titers) vaccinated with D614 monovalent vaccine, all vaccinated hamsters, whatever the vaccine formulations, had stable or slightly increasing body weight post-challenge, following the same profile as the non-challenged group.

Viral loads were measured in lungs 4- and 7-days post-challenge (half of the animals per group and per timepoint) using 50% tissue culture infectious dose (TCID$_{50}$) titration. In each cohort, the unvaccinated group (buffer) showed high viral loads 4 days post-challenge at mean viral loads of 8.8 log$_{10}$, 8.6 log$_{10}$, and 8.3 log$_{10}$ TCID$_{50}$ in the D614G, Alpha, and Beta cohorts, respectively (Fig. 8b). On D4 post-challenge, hamsters vaccinated with the bivalent or the monovalent Beta vaccines displayed low to undetectable viral loads, irrespectively of the virus used for the challenge (D614G, Alpha, or Beta). Hamsters vaccinated with the

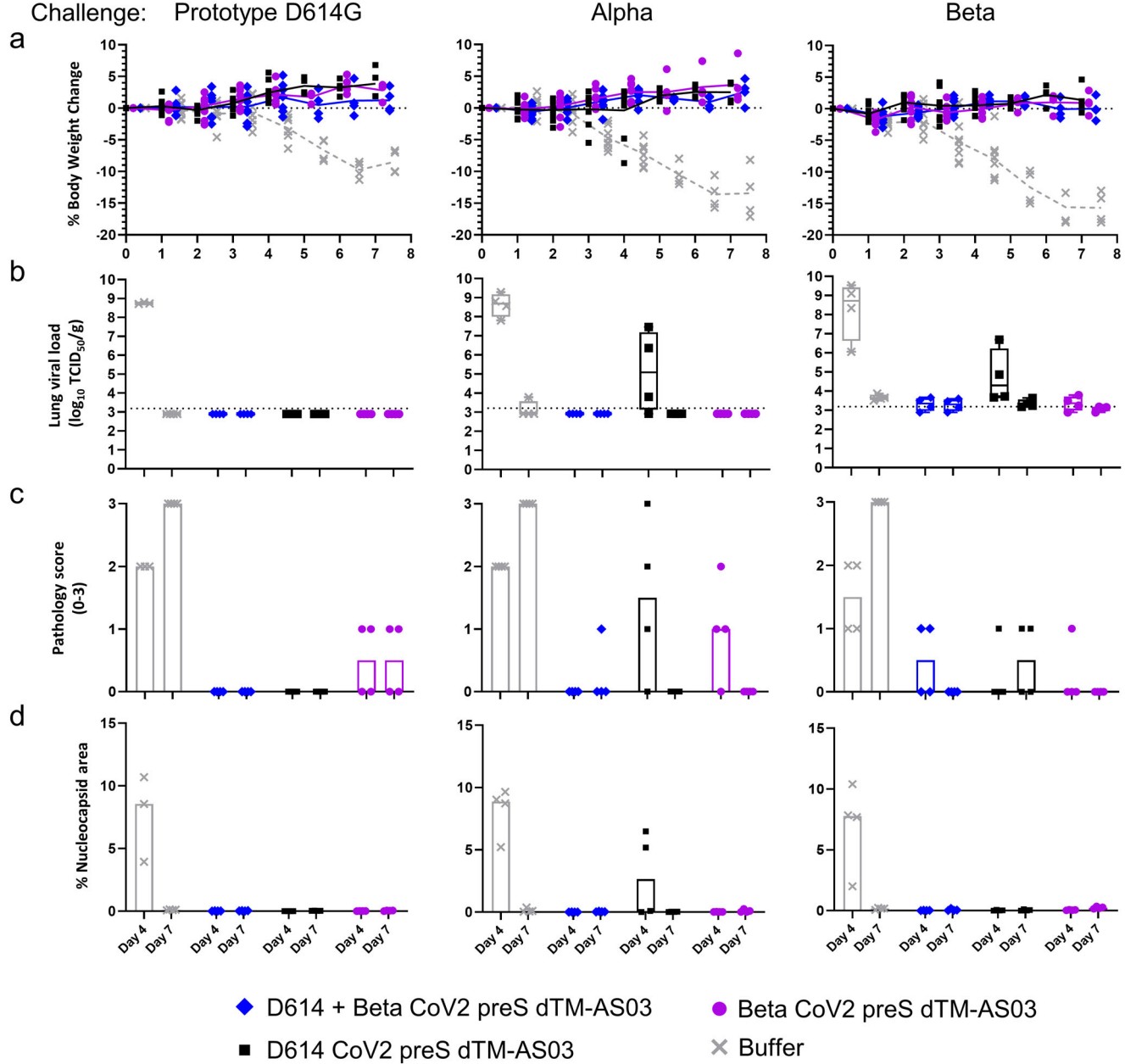

**Fig. 8 Bivalent CoV2 preS dTM-AS03 (D614 + Beta) vaccine efficacy in hamsters after prototype D614G, Alpha, or Beta virus challenge.** Hamsters vaccinated with the bivalent D614 + Beta (1 μg + 1 μg) and the monovalent D614 or Beta at (1 μg), were challenged (N = 8/group) with prototype D614G, Alpha, or Beta viruses, 28 days after the second dose. **a** Individual body weight change was assessed daily, during the 7 days following challenge. Symbols represent individual data and lines the mean of the group. **b** Viral load by infectious titration, **c** pathology score (based on % of tissue affected: 0 = 0%; 1 < 25%; 2 = 25–50%; 3 > 50%), and **d** nucleocapsid protein area percentage were assessed in the lungs of individual hamster, 4- or 7-days post-challenge. Bars represent the median of the group for pathology score and nucleocapsid protein area percentage, the dashed line represents the limit of detection.

monovalent D614 vaccine also displayed low to undetectable viral loads on D4, except for the two hamsters in the Alpha challenge cohort that showed high viral loads consistent with body weight loss and no detectable nAb titers. One hamster vaccinated with the monovalent D614 also showed a high viral load in the cohort challenged with the Beta variant.

In the three cohorts, infectious titers were low to undetectable 7 days post-challenge.

We next assessed lung pathology 4- and 7-days post-challenge by microscopic evaluation of lung sections stained with hematoxylin and eosin (H&E) or subjected to anti-SARS-CoV-2 nucleocapsid protein immunohistochemistry (IHC). For all three viral strains, pathology in unvaccinated hamsters (buffer)

consisted of multifocal to coalescing regions of inflammatory infiltrate composed of macrophages, lymphocytes, heterophils, and syncytial cells admixed with hemorrhage and cellular debris often obliterating normal architecture (Fig. 9a and Supplementary Fig. 5). There was marked type II pneumocyte hyperplasia and bronchiolar epithelium was multifocally hyperplastic characterized by many epithelial cells piling up and interspersed with rare necrotic cells. The lumena of some bronchioles contained necrotic cellular debris. Blood vessels reveal either a mural or perivascular infiltration by mixed inflammatory infiltrates. Lung pathology was markedly attenuated in hamsters that received the vaccination. Semi-quantitative scores were made based on the percent of lung affected.

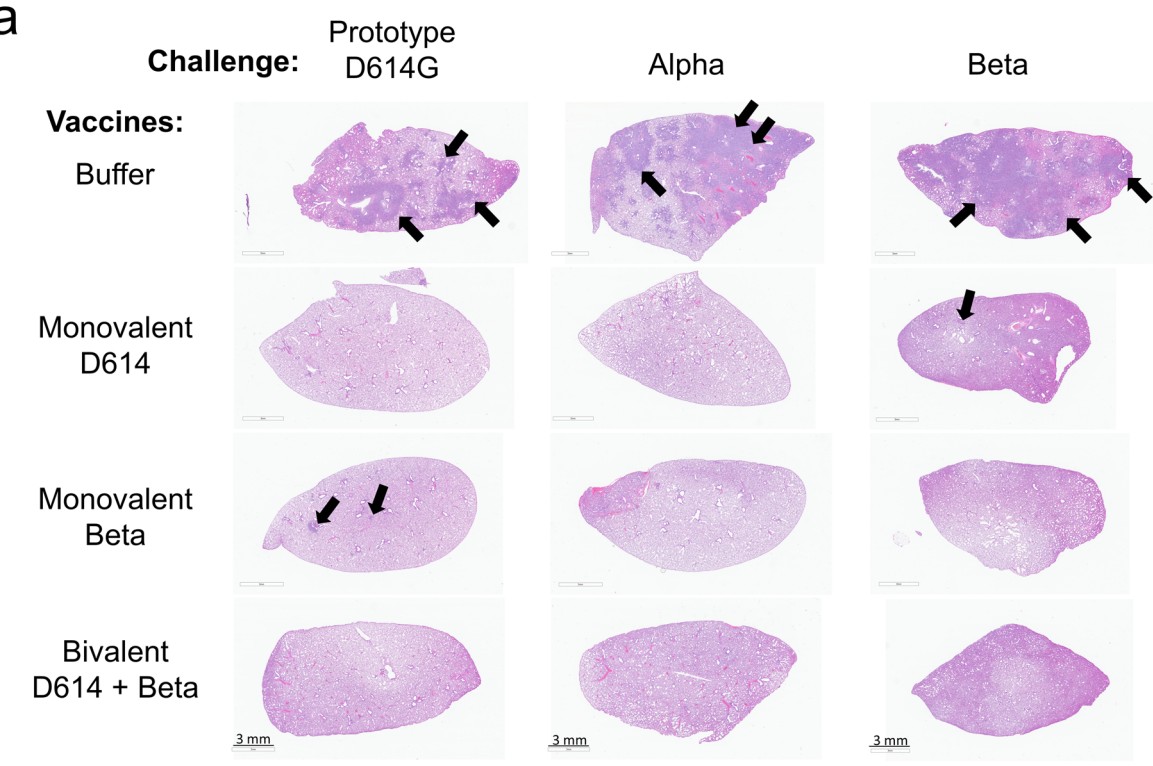

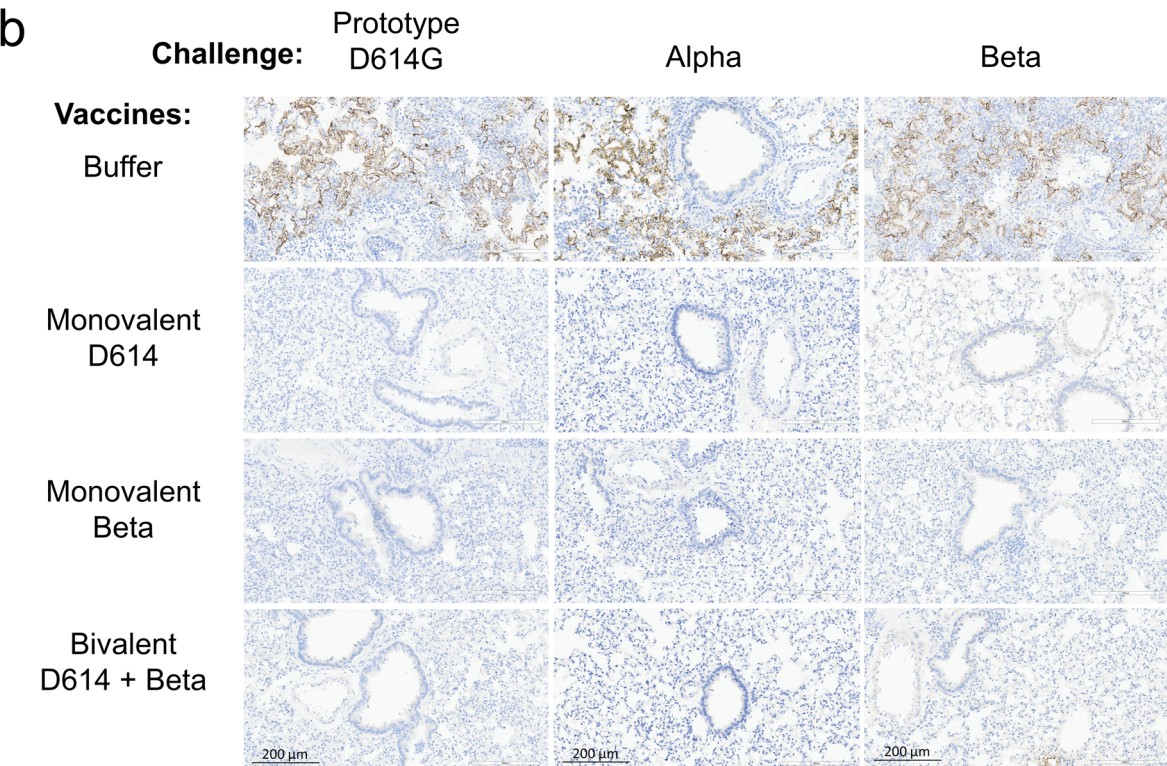

**Fig. 9 Bivalent CoV2 preS dTM-AS03 (D614 + Beta) vaccine prevents lung histopathology induced by SARS-CoV-2 VOC in hamsters. a** Representative low magnification (8x) photomicrographs (H&E). The lungs from hamsters receiving buffer only, the monovalent D614 or Beta (1 μg) and the bivalent D614 + Beta (1 μg + 1 μg) vaccines were assessed after H&E staining, 7 days post-challenge with prototype D614G, Alpha, and Beta viruses. Arrows = inflammatory infiltrate, type II pneumocyte hyperplasia, cellular debris represented by dark purple regions. **b** Nucleocapsid protein expression was analyzed by immunohistochemistry in the lungs of hamsters 4 days post-challenge with prototype D614G, Alpha, and Beta viruses.

All unvaccinated hamsters had median lung pathology scores of 2 (on a scale of 0–3), 4 days post-challenge with prototype D614G and Alpha, and a score of 1 or 2 in the group challenged with Beta (Fig. 8c). Seven days post-challenge, the pathology progressed in all unvaccinated animals to the maximal score of 3, irrespective of the strain used for the challenge. In contrast, hamsters vaccinated with the monovalent or bivalent vaccines displayed no or minimal pathology (scores of 0 or 1), at day 4 or 7 post-challenge, except for the two low responder hamsters vaccinated with D614 and challenged with the Alpha variant, which displayed scores of 2 and 3, consistent with the high viral loads. In the same cohort (Alpha challenge), one hamster vaccinated with the monovalent Beta showed a score of 2, 4 days post-challenge; however, post hoc analysis indicated that the pathology was less developed than in the buffer immunized controls with similar scores. Moreover, while immunohistochemistry showed nucleocapsid protein expression in multifocal to coalescing regions of the lungs in unvaccinated hamsters 4 days post-challenge, nucleocapsid protein was not detected in any vaccinated hamsters at the same timepoint, except in the two low responder animals immunized with the monovalent D614 vaccine (Figs. 8d, 9b).

Overall, the D614/Beta bivalent, as well as the two monovalent CoV2 preS dTM-AS03 (D614 and Beta) vaccines, conferred protection in hamsters against body weight loss, viral replication in lungs and lung pathology induced by prototype D614G, Alpha, or Beta variants.

## Discussion

SARS-CoV-2 transmission is still uncontrolled in many parts of the world, and the emergence of vaccine-resistant variants such as Omicron and its subvariants have highlighted the limitations of vaccine booster strategies based on the ancestral D614 Wuhan strain and stimulated the development of new vaccine booster formulations based on variant spikes[32–36].

Here, we evaluated the immunogenicity and efficacy conferred by a bivalent recombinant CoV2 preS dTM vaccine formulation with AS03-adjuvant containing the prefusion-stabilized Spike trimers of the ancestral D614 and the Beta (B.1.351) variant strains in naïve NHPs and hamsters after a primary immunization series. The NHP model has high predictivity for COVID-19 vaccine immunogenicity and the Golden Syrian Hamster has proven to be a model of choice to assess vaccine efficacy against pathology caused by SARS-CoV-2 variants[37,38].

Findings from our study demonstrate the broad and durable immunogenicity conferred by the D614/Beta bivalent protein-based CoV2 preS dTM-AS03 vaccine in NHPs up to 1 year. Specifically, the bivalent vaccine induces high neutralizing antibody responses against the vaccine-homologous viruses (D614, D614G and Beta) and Alpha, Gamma, Delta, and Mu VOC, reaching stable levels between 3 and 12 months. Importantly, the bivalent vaccine also elicits cross-neutralizing antibodies against SARS-CoV-1 from the 2003 outbreak as well as consistent and persistent cross-neutralizing antibodies against Omicron BA.1 and BA.4/5. We also measured the induction of high levels of Spike-specific memory B cells (ancestral D614 and Omicron BA.4/5) at 1-year post-immunization with the three formulations. Furthermore, the combination of the two antigens (D614 and Beta CoV2 preS dTM) in the bivalent vaccine formulation did not reduce the immunogenicity of either antigen when compared to the monovalent formulations. The extensive cross-neutralizing responses induced by the bivalent formulation can be explained by the additive effects of the two vaccine components, which display opposite profiles with regard to the neutralization of variants. Indeed, the D614 monovalent vaccine induces cross-neutralizing Abs against D614G, Alpha, and Delta containing the original E484 amino acid, while the Beta monovalent vaccine induces cross-neutralizing Abs against Gamma and Mu containing the E484K mutation responsible for most of the antibody escape[20]. The consistent neutralization of the much more distant Omicron variant (with E484A mutation) could be the result of a complementary or synergistic effect towards conserved epitopes that might be subdominant in each monovalent vaccine[39,40].

To complement the neutralization results, extensive system serology analyses indicated that the bivalent formulation elicited overall similar IgG subclasses and Fc functions as compared to each monovalent formulation.

Interestingly, when we compared the two-dose immunization regimen with the bivalent vaccine to alternative heterologous immunization regimens, with monovalent D614 followed by monovalent Beta or bivalent vaccines, only the second injection with monovalent Beta induced balanced neutralizing responses, and with a higher variability than two-doses with the bivalent vaccine. The lower performance of the heterologous prime/boost was also observed in mice where lower nAbs (ancestral, Alpha, Beta, and Gamma) were induced with heterologous vaccination with S-trimer vaccine (D614 on D0 and Beta on D21), compared to two-doses with a bivalent vaccine[41]. The limited breadth of responses observed in our study with the D614/Beta heterologous regimen contrasts with the expanded breadth of neutralizing responses observed recently after a late booster vaccine in NHPs and humans[14–16]. This likely relates to the absence of maturation of the memory B cell population when the second injection is performed shortly after the priming immunization (3 weeks here compared to 6 months to 1 year in the context of a booster vaccination)[42,43]. While in the heterologous D614 prime/Bivalent boost regimen, the D614 Ab profile observed corresponds to the "original antigenic sin", where the first exposure determines the Ab responses elicited by booster vaccinations[44,45], in the context of the COVID booster vaccination, the primary immunization efficiently activate germinal centers which then generate broader Ab responses with time[46].

The study showed stability of the neutralizing titers between 3 to 12 months with the different CoV2 preS dTM-AS03 vaccine formulations after a two-dose primary series. This contrasts with the continuous nAb decline observed with the mRNA-1273 vaccine over a similar period (up to D209) in humans[47].

To our knowledge, these are the first data showing extensive cross-neutralization covering all emerging VOC up to 12 months in NHPs after a two-dose primary immunization. Prior studies showed that bivalent-based or nanoparticle-based vaccine candidates were able to induce balanced neutralizing Ab responses against VOC, but the data were limited to mice or documented on a more limited panel of variants[41,48–50]. Other candidates, based on Spike-ferritin nanoparticles or RBD-ferritin nanoparticles, showed similar breadth against SARS-CoV-1 and Ab durability; however, these candidates are at earlier clinical development stages[51–53]. Future studies on long-lived plasma cells and memory B cell compartments would be warranted to better understand the durability of the Ab responses with mRNA and protein-based vaccines.

In this study, we also showed the vaccine efficacy in hamsters where the Beta-containing bivalent vaccine formulation protected from lung pathology and viral replication after a challenge with the D614G, Alpha, and Beta variants. The protection against infection and pathology was observed in animals with reduced cross-neutralizing antibody titers elicited by immunization with the monovalent formulations, supporting the absence of enhanced disease in the context of heterologous infection. Of note, two hamsters in the monovalent D614 vaccine group showed low antibody binding titers (ELISA) and no detectable

neutralizing antibody titers, 2 weeks post-second dose. These observations correlated with clinical signs of infection after the challenge (weight loss, viral replication, and lung pathology). An exploratory analysis indicated that protection against pathology was associated with detectable neutralizing antibody titers; however, no threshold could be identified. Future studies would be needed to define thresholds of protection for variants[54,55] and to explore the role of vaccine-elicited cross-reactive T cell responses in vaccine protection[56].

Although the studies presented here involved a low number of animals per group and assessed protection shortly after immunization, the results were recently confirmed in a large efficacy clinical trial (NCT04904549), where high levels of protection against Omicron symptomatic infection were demonstrated after primary immunization with the Beta-containing bivalent CoV2 preS dTM-AS03 vaccine[57]. Interestingly, in two other Phase 3 clinical trial (NCT04762680, NCT05124171), the Beta monovalent CoV2 preS dTM-AS03 vaccine showed a significant superiority to the D614-based vaccine in boosting Omicron nAbs in adults previously primed with mRNA COVID-19 vaccines[16,17].

Considering the continuous rapid viral evolution selecting for immune escape variants, both the naïve population, such as the young or those not yet vaccinated, and the previously vaccinated population, might benefit from the broad and durable immune responses conferred by the Beta-containing CoV2 preS dTM-AS03 vaccine formulations.

## Data availability

Protein sequences are available on GISAID using the accession codes provided in the manuscript: B.1.351 sequences GISAID Accession EPI_ISL_1048524. The source data generated in this study are provided as Supplementary Data 1. All other data were available from the corresponding author (or other sources, as applicable) on reasonable request.

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

## Acknowledgements

Histopathology and IHC team members of Sanofi Global Discovery Pathology (TIM) for their excellent technical help in histology and IHC procedures on the hamster lung samples.The authors thank Jon Smith for coordinating the production and providing the vaccine antigens for the study, Caroline Patriarca-Ruat for project management, and Carlos Diaz-Granados, Stephen Savarino, and Saranya Sridhar for critical discussions on the study designs and data analysis. The authors thank Dean Huang for testing hamster sera in ELISA and Julie Piolat for statistical analyses support. The authors also thank Isabel Grégoire, Hardik Ashar, Priya Upadhyay, and Hanson Geevarghese (Sanofi) for providing editorial assistance and paper coordination. This work was done in collaboration with GSK, who provided access to and use of the AS03 adjuvant system. Funding was provided in part by Sanofi, and from the Biomedical Advanced Research and Development Authority (BARDA), Administration for Strategic Preparedness and Response at the US Department of Health and Human Services under Contract # HHSO100201600005I, and in collaboration with the US Department of Defense Joint Program Executive Office for Chemical, Biological, Radiological and Nuclear Defense under Contract # W15QKN-16-9-1002.

## Author contributions

V.L., T.T., C.B., V.P., A.R., S.G., N.G.A., R.M.C., C.G., M.Ko, C.A., and G.A. contributed to the concept or design of the study. M.K., S.G., A.R., S.G., D.S.B., S.K., and C.A. contributed to performing the study and sample analysis. V.L., T.T., C.B., V.P., A.R., S.G., N.G.A., R.M.C, C.G., M.K., C.A., G.A., C.G., and M.Ko were involved in the interpretation of the data and review of the manuscript. V.L., V.P., and C.B. drafted the first manuscript.

## Competing interests

C.B., V.P., N.G.A., M.K., D.H., T.T., A.R., S.G., D.S.B., R.M.C., and V.L. are employees of the Sanofi company and may hold shares. S.C. was an employee of Sanofi at the time of study conduct and currently employed with AbbVie and holds shares of Sanofi. S.K. is an employee of Bioqual and reports no conflicts. G.A. is a cofounder and serves as a consultant to SeromYx Systems Inc and has a patent pending through SeromYx Systems Inc. C.A. and G.A. were employees of Ragon Institute of MGH, MIT, and Harvard at the time of study conduct and currently employed with Moderna. Y.D. has nothing to disclose. M.Ko and C.G are employees of the GSK group of companies and report ownership of GSK shares.
