## [Peer Review File · Communications Medicine]

Reviewers' comments:

Reviewer #1 (Remarks to the Author):

The manuscript submission by Valerie Lecouturier and colleagues details findings from preclinical assessment of an AS03-adjuvanted COVID-19 bivalent vaccine containing pre-fusion spike of parental and Beta (B.1.351) strains of SARS-CoV-2. The authors assessed immunogenicity and protective efficacy in macaque and hamster models of COVID-19. The results indicate the bivalent vaccine is safe and immunogenic with negligible antigen interference. Sequential immunization at day 0 and day 21 induced broad antibody responses that neutralized the parental D614 strain as well as variants of concern. The abbreviated vaccination interval presents a minor concern for lack of recall of memory responses. Another minor concern pertains to the pathology scores detailed in Figure 7 in which the some of the pathology scores, notably for Alpha, are not consistent with the lung viral loads. In addition, lines 674 to 678 should be removed as this repeats information provided in the Methods section. In general, the results reflect current clinical practice that was recently updated to include a similar bivalent COVID-19 vaccination booster.

Reviewer #2 (Remarks to the Author):

"A Beta-containing bivalent SARS-CoV-2 spike protein vaccine candidate with AS03 elicits durable and broad neutralization of variants including Omicron in macaques and confers protection in hamsters" try to address the question of whether primary immunization with the bivalent and adjuvanted CoV2 preS vaccine can induce broader and durable neutralizing antibody responses against VOC including Omicron BA.1, and SARS-CoV-1 as compared to the parental D614 or Beta variant monovalent vaccines in NHP. Additionally, authors performed challenge experiments in mice to conclude that the bivalent vaccine conferred protection against lung pathology and viral replication induced after challenge with the parental virus and the Alpha and Beta variants.

Generally, the manuscript has potential and is supported by the decent number of animals included. However, manuscript lacks clarity, figures are confusing and methods for some assays are not explained in detail. Finally, some conclusions, such as protection against Omicron or SARS-CoV-1 conferred by the bivalent vaccine are overstated.

Specific comments:

Reviewer would appreciate if authors could, at least briefly, describe the vaccine used in the study, e.g. in the material and method section. Line 108: "Vaccine candidates CoV2 preS dTM D614 or Beta with AS03-adjuvant and formulated as monovalent or bivalent vaccines were described previously (14, 21)". Info is very little and would be appropriate to include some details, so the reader does not have to go to original manuscripts Francica JR, et al. or Pavot V, et al.

Line 143: "They were vaccinated on D0 and D21 by intramuscular route into the deltoid muscle with 3 vaccine formulations: monovalent D614 or Beta at 5µg or 10µg or bivalent (D614 + Beta) at 5 µg + 5 µg. Two other groups received heterologous D0/D21 vaccines composed of monovalent D614 at 5 µg on D0 and either monovalent Beta at 5 µg or bivalent D614 + Beta at 5 µg + 5 µg on D21" It is a bit confusing what vaccine receive each group when one reads the methods of the manuscript for the first time. Authors should refer to the figure explaining the vaccination regimen. Same for vaccination description in hamsters.

Why authors challenge the hamsters with such a different number of viruses?? I understand this is due to the intrinsic pathogenicity of each variant, and according to the weight loss, but I wonder if one should then plan the experiments and include different dilutions of each variant and show the comparisons or perhaps a score similar to LD50. Please justify.

Reviewer would appreciate if authors could, at least briefly, describe the convalescents human sera panels used without have to look for the original publication (line 162). Similar for the assays performed for the pseudo neutralization (line 180) or ELISA assays (line 184).

Figure 2-3, mean values of antibodies are geometric mean titers? If not, data should be expressed in GMT, as per convention. Similarly, some type of error bars such as CI95% should be shown.

Line 427: "For comparison, the bivalent vaccine elicited mean D614 nAb titers higher than that measured in the World Health Organization (WHO) International Standard for anti-SARS-CoV-2 immunoglobulin" Please explain the standard, is a bit confusing.

Line 450: "Among the two heterologous prime/boost regimens, that with the monovalent Beta used for the second injection was able to extend the breadth of the neutralizing response to Beta and other VOC" Can authors explain to compared to what?? The first vaccination series included in Figure 2, or within groups on Figure S2? Additionally, please explain doses used for these experiments in the text, not only the figure legend.

Figure 3D: Not clear if authors were not able to detect any titers at day 70 and 114 or assays were not performed. If the first, how do you explain that authors are able to detect at day 206???

Line 487: "D614G and Beta variant mean nAb titers at 6 months were...." Per convention titers should be expressed as GMT and errors (95% CI) should be given. Please harmonize.

Figure 5C: the bivalent formulation induced significantly less antibody-dependent cellular phagocytosis (ADCP) against D614 Spike compared to the monovalent formulation at 10 µg, what's the biological meaning of this, is bivalent better or worse than monovalent?? Differences in antibody mediated functions deserve some discussion.

Figure S4 B-C: what is meant by the "low" in the Y axis title?? Also, Fig 5 and S4 show lineages of SARS-CoV-2 VOCS while the other figures shown convention names of the variants, e.g. B.1.351 or Beta. Please harmonize.

Line 665: nucleoprotein should be instead nucleocapsid.

Line 721: timers should read trimers.

Line 729: "Importantly, the bivalent vaccine also elicits consistent and persistent cross-neutralizing antibodies against Omicron BA.1 and SARS-CoV-1 from the 2003 outbreak" I disagree with this statement, authors did not show consistent pattern here for Omicron or SARS-CoV-1, nor protection in hamsters after challenge.

Response to Reviewers

Reviewer #1 (Remarks to the Author):

The manuscript submission by Valerie Lecouturier and colleagues details findings from preclinical assessment of an AS03-adjuvanted COVID-19 bivalent vaccine containing pre-fusion spike of parental and Beta (B.1.351) strains of SARS-CoV-2. The authors assessed immunogenicity and protective efficacy in macaque and hamster models of COVID-19. The results indicate the bivalent vaccine is safe and immunogenic with negligible antigen interference. Sequential immunization at day 0 and day 21 induced broad antibody responses that neutralized the parental D614 strain as well as variants of concern.

The abbreviated vaccination interval presents a minor concern for lack of recall of memory responses.

Response: We thank the reviewer for this comment, and we agree that the short interval (3 weeks) between dose 1 and dose 2 may have an impact on memory responses. This immunization regimen was selected early during the pandemic, due to the emergency context, and has been used since in all non-clinical and clinical primary immunization schedule. Although probably not optimal, using this immunization regimen, we show a good longevity of the circulating neutralizing Abs (for at least 206 days), as well as strong induction of spike-specific memory B cells that are able to be reactivated upon antigen exposure in our study, and by a third vaccination in an independent study in non-human primates (Pavot et al, Nat Communications, 2022, <https://doi.org/10.1038/s41467-022-29219-2>).

Another minor concern pertains to the pathology scores detailed in Figure 7 in which some of the pathology scores, notably for Alpha, are not consistent with the lung viral loads.

Response: Pathology scores depicted in Figure 7c compare the effect on lung pathology of a challenge with the prototype D614G strain or the alpha or beta variants in hamsters vaccinated with either the bivalent vaccine, monovalent vaccines or buffer (mock control). In this acute virus infection model, in non-vaccinated animals, lung viral load peaks at day 4 while the pathology changes continue to worsen up to day 7 (due to reparative inflammation, epithelial regeneration and fibroplasia), although viral loads are undetectable or minimal on Day 7. The differential kinetics of viral load and lung pathology changes as well as a few non-responder animals in some study cohorts may explain the apparent lack of concordance between viral load and lung pathology scores. Note that 2 out of the 3 high scores (>2) in the D614-vaccine group from the Alpha challenge cohort correspond to the 2 non-responder animals, displaying high viral loads and nucleocapsid staining. If we exclude these 2 non-responder animals, the pathology scores in Alpha cohort are 1 score of 2 and 4 scores of 1, compared to 4 scores of 1 in the D614G cohort and 6 scores of 1 in the Beta cohorts, which is comparable.

Importantly, in all vaccinated cohorts, lung pathology at day 4 was lower than that in mock-vaccinated control and seems to further ameliorate at day 7 contrasting with the worsening observed in control animals, indicating clear vaccine efficacy in minimizing viral load in the lung on day 4, rapid clearance by day 7 as well as a reducing and controlling lung pathology. Although a few poor responders (hamsters in Alpha challenged, D614 monovalent vaccinated) do seem to have

higher lung pathology at day 7 that coincides with higher viral load; both viral loads and lung pathology are ameliorated on day 7. Overall, in context of this study and despite the low numbers of animals per treatment cohort, these data show clear efficacy of the bivalent vaccine in preventing viral infection and ameliorating the lung pathology.

In addition, lines 674 to 678 should be removed as this repeats information provided in the Methods section.

Response: we removed the lines accordingly.

In general, the results reflect current clinical practice that was recently updated to include a similar bivalent COVID-19 vaccination booster.

Reviewer #2 (Remarks to the Author):

A Beta-containing bivalent SARS-CoV-2 spike protein vaccine candidate with AS03 elicits durable and broad neutralization of variants including Omicron in macaques and confers protection in hamsters” try to address the question of whether primary immunization with the bivalent and adjuvanted CoV2 preS vaccine can induce broader and durable neutralizing antibody responses against VOC including Omicron BA.1, and SARS-CoV-1 as compared to the parental D614 or Beta variant monovalent vaccines in NHP. Additionally, authors performed challenge experiments in mice to conclude that the bivalent vaccine conferred protection against lung pathology and viral replication induced after challenge with the parental virus and the Alpha and Beta variants. Generally, the manuscript has potential and is supported by the decent number of animals included.

However, manuscript lacks clarity, figures are confusing and methods for some assays are not explained in detail.

Finally, some conclusions, such as protection against Omicron or SARS-CoV-1 conferred by the bivalent vaccine are overstated.

Specific comments:

Reviewer would appreciate if authors could, at least briefly, describe the vaccine used in the study, e.g. in the material and method section. Line 108: “Vaccine candidates CoV2 preS dTM D614 or Beta with AS03-adjuvant and formulated as monovalent or bivalent vaccines were described previously (14, 21)”. Info is very little and would be appropriate to include some details, so the reader does not have to go to original manuscripts Francica JR, et al. or Pavot V, et al.

Response: We added that information in the Material & Methods lines 111-118.

CoV2 preS dTM vaccine candidates consist of a stabilized prefusion trimeric recombinant SARS-CoV-2 S protein. The CoV2 preS dTM (D614) and Beta were designed based on the Wuhan YP_009724390.1 and B.1.351 sequences (GISAID Accession EPI_ISL_1048524), respectively, with 2 prolines in the S2 domain, deletion of the transmembrane region, and addition of the T4 foldon trimerization domain. The CoV2 preS dTM D614 or Beta were produced using a Sanofi proprietary cell culture technology based on the insect cell-baculovirus expression system. The CoV2 preS dTM D614 or Beta are formulated with the AS03 adjuvant from GSK.

Line 143: “They were vaccinated on D0 and D21 by intramuscular route into the deltoid muscle with 3 vaccine formulations: monovalent D614 or Beta at 5µg or 10µg or bivalent (D614 + Beta) at 5 µg + 5 µg. Two other groups received heterologous D0/D21 vaccines composed of monovalent D614 at 5 µg on D0 and either monovalent Beta at 5 µg or bivalent D614 + Beta at 5 µg + 5 µg on D21” It is a bit confusing what vaccine receive each group when one reads the methods of the manuscript for the first time. Authors should refer to the figure explaining the vaccination regimen. Same for vaccination description in hamsters.

Response: The text has been changed to clarify the different vaccine groups (lines 143-149).

“Macaques were vaccinated on D0 and D21 by intramuscular route into the deltoid muscle with different vaccine formulations: monovalent ancestral spike D614 (at 5µg or 10µg), monovalent Beta variant spike (at 5µg or 10µg) or bivalent D614 + Beta (at 5 µg + 5 µg) (Fig. 1). Two other groups received the monovalent D614 at 5 µg on D0 and a heterologous vaccine on D21 composed of either monovalent Beta at 5 µg or bivalent D614 + Beta at 5 µg + 5 µg (Supp Fig. 2a).”

Why authors challenge the hamsters with such a different number of viruses?? I understand this is due to the intrinsic pathogenicity of each variant, and according to the weight loss, but I wonder if one should then plan the experiments and include different dilutions of each variant and show the comparisons or perhaps a score similar to LD50. Please justify.

Response: Indeed, the different doses used are due to the different intrinsic pathogenicity of the variants. During the development of the hamster model, we selected non-lethal doses of SARS-CoV-2 viruses, different dilutions (and thus doses) were tested for body weight loss. Hence, LD50 value can't be determined from the preliminary experiments. For each variant, the infectious doses inducing comparable weight loss around 15-18% post challenge were selected. However, some stocks, like Alpha are less pathogenic than others, so higher doses were used. Titration data from these stocks could be shared if required.

Reviewer would appreciate if authors could, at least briefly, describe the convalescents human sera panels used without have to look for the original publication (line 162). Similar for the assays performed for the pseudo neutralization (line 180) or ELISA assays (line 184).

Response: this was done accordingly (lines 165-169, lines 176-189 and lines 192-205).

Figure 2-3, mean values of antibodies are geometric mean titers? If not, data should be expressed in GMT, as per convention. Similarly, some type of error bars such as CI95% should be shown.

Mean values were calculated on titers expressed in \log_{10} , we thus used the arithmetic mean. The arithmetic mean of logarithmic values is equivalent to geometric mean of arithmetic values. CI95% error bars have been added on the graph.

Line 427: "For comparison, the bivalent vaccine elicited mean D614 nAb titers higher than that measured in the World Health Organization (WHO) International Standard for anti-SARS-CoV-2 immunoglobulin" Please explain the standard, is a bit confusing.

Response: We added an explanation line 451-455 explaining that the International Standard for anti-SARS-CoV-2 immunoglobulins were adopted by the WHO Expert Committee on Biological Standardization on Dec 10, 2020. The International Standard allows the accurate calibration of assays to an arbitrary unit, thereby reducing inter-laboratory variation and creating a common language for reporting data (ref: [https://www.thelancet.com/journals/lancet/article/PIIS0140-6736\(21\)00527-4/fulltext](https://www.thelancet.com/journals/lancet/article/PIIS0140-6736(21)00527-4/fulltext)).

Line 450: "Among the two heterologous prime/boost regimens, that with the monovalent Beta used for the second injection was able to extend the breadth of the neutralizing response to Beta and other VOC" Can authors explain to compared to what?? The first vaccination series included in Figure 2, or within groups on Figure S2? Additionally, please explain doses used for these experiments in the text, not only the figure legend.

Response:

The text was changed as follows:

Compared to the monovalent D614 vaccine 2-dose regimen, among the two heterologous prime/boost regimens, that with the second injection with monovalent Beta used for the second injection was able to extend the breadth of the neutralizing response to Beta and other VOC, while the second injection with the bivalent didn't improve the breadth. However, the heterologous prime/boost with monovalent Beta for the second injection induced although with higher variability and slightly lower titers than the bivalent vaccine 2-dose regimen (not significant for D614, D614G, Alpha, and Delta mean nAb titers, 3- and 4.1-fold lower for Beta, $p=0.0165$ and Gamma, $P = 0.0052$ mean nAb titers, respectively) (Supplementary Fig. 2b).

Figure 3D: Not clear if authors were not able to detect any titers at day 70 and 114 or assays were not performed. If the first, how do you explain that authors are able to detect at day 206???

Response: the assays were not performed at D70 and D114 for Omicron. We added this information Line 509.

We also changed the figure 3 since we generated new data at 1 year (D365). We added this time point to D614G, Beta and Omicron BA.1 graphs.

We moved neutralizing titers against Delta in the Supp Fig 3 since this variant is less relevant now.

We also added the Figure 4 showing neutralizing titers against Omicron BA.4/5 at 1 year.

As we generated new spike-memory B-cell data at 1 year (against Ancestral and Omicron BA.5) we replaced the memory B cells at D114 by memory B-cells at 1 year post-immunization (new Fig. 5).

Line 487: "D614G and Beta variant mean nAb titers at 6 months were...." Per convention titers should be expressed as GMT and errors (95% CI) should be given. Please harmonize.

Response: We agree that when using arithmetic numbers, mean titers are calculated with the geometric mean and expressed as GMTs. In this study, we used \log_{10} values for which we can calculate the means using the arithmetic mean function, which is equivalent to the GMT of arithmetic numbers. The conclusions are thus identical.

Figure 5C: the bivalent formulation induced significantly less antibody-dependent cellular phagocytosis (ADCP) against D614 Spike compared to the monovalent formulation at 10 μg , what's the biological meaning of this, is bivalent better or worse than monovalent?? Differences in antibody mediated functions deserve some discussion.

Response: While the bivalent formulation induced significantly less ADCP against D614 compared to the higher-dose D614 monovalent approach, this was not consistent across VOC. In fact, no significant difference was noted in the higher dose B.1.351 monovalent compared to the bivalent vaccine for ADCP, ADCD, or ADNP against all other variants tested, including Omicron. We therefore conclude that the observed increase in ADCP against D614 due to the higher dose of the monovalent vaccine was not consistent across VOC. Rather, we conclude that a bivalent, Beta-Spike containing vaccine can elicit a pan-VOC, multi-functional antibody response that is comparable to antigen-specific monovalent vaccines, even when the latter is administered at higher doses.

Figure S4 B-C: what is meant by the "low" in the Y axis title?? Also, Fig 5 and S4 show lineages of SARS-CoV-2 VOCS while the other figures shown convention names of the variants, e.g. B.1.351 or

Beta. Please harmonize.

Response: We performed the assay with multiple dilutions for the same antigen shown in Supplementary 4 B-C is the representative data. We have made a reference to this in the Luminex subsection of the methods. We have eliminated the term “low” on the y-axis in Supp Figure 4 to avoid confusion.

Line 665: nucleoprotein should be instead nucleocapsid.

Response: For coronaviruses, nucleocapsid protein is the term used (Perlman & Dandekar, Nat Reviews Microb, 2005), this has been harmonized accordingly.

Line 721: timers should read trimers.

Response: this has been corrected accordingly.

Line 729: “Importantly, the bivalent vaccine also elicits consistent and persistent cross-neutralizing antibodies against Omicron BA.1 and SARS-CoV-1 from the 2003 outbreak” I disagree with this statement, authors did not show consistent pattern here for Omicron or SARS-CoV-1, nor protection in hamsters after challenge.

Response: we changed the phrasing to make the conclusions clearer based on our data. Indeed, in NHP immunized with the bivalent vaccine, we measured cross-neutralizing titers against SARS-CoV-1 at 2 weeks post-dose 2, 6 months and one year (Fig 3d). At all these timepoints, we observed cross-neutralizing titers against Omicron BA.1 at 2 weeks post-dose 2 and those titers were still detectable at 6 months in all macaques immunized with the bivalent vaccine.

REVIEWERS' COMMENTS:

Reviewer #1 (Remarks to the Author):

The reviewer appreciates the effort of the authors to revise the manuscript. The authors have satisfactorily addressed the concerns.

Reviewer #2 (Remarks to the Author):

Authors have addressed most of the queries, except for some details, like harmonization of VOC used, either lineages or convention names should be harmonized or at least referred to in the legend of Fig S4 and Fig 5.